# Differential Fairness: An Intersectional Framework for Fair AI [note 1]

**DOI:** 10.3390/e25040660

**Published:** 2023-04-14

**Authors:** Rashidul Islam, Kamrun Naher Keya, Shimei Pan, Anand D. Sarwate, James R. Foulds

**Affiliations:** 1Department of Information Systems, University of Maryland, Baltimore County, Baltimore, MD 21250, USA; islam.rashidul@umbc.edu (R.I.); kkeya1@umbc.edu (K.N.K.); shimei@umbc.edu (S.P.); 2Department of Electrical and Computer Engineering, Rutgers, The State University of New Jersey, New Brunswick, NJ 08854, USA; anand.sarwate@rutgers.edu

**Keywords:** fairness in AI, AI and society, intersectionality, 80% rule, privacy

## Abstract

We propose definitions of fairness in machine learning and artificial intelligence systems that are informed by the framework of intersectionality, a critical lens from the legal, social science, and humanities literature which analyzes how interlocking systems of power and oppression affect individuals along overlapping dimensions including gender, race, sexual orientation, class, and disability. We show that our criteria behave sensibly for any subset of the set of protected attributes, and we prove economic, privacy, and generalization guarantees. Our theoretical results show that our criteria meaningfully operationalize AI fairness in terms of real-world harms, making the measurements interpretable in a manner analogous to differential privacy. We provide a simple learning algorithm using deterministic gradient methods, which respects our intersectional fairness criteria. The measurement of fairness becomes statistically challenging in the minibatch setting due to data sparsity, which increases rapidly in the number of protected attributes and in the values per protected attribute. To address this, we further develop a practical learning algorithm using stochastic gradient methods which incorporates stochastic estimation of the intersectional fairness criteria on minibatches to scale up to big data. Case studies on census data, the COMPAS criminal recidivism dataset, the HHP hospitalization data, and a loan application dataset from HMDA demonstrate the utility of our methods.

## 1. Introduction

The increasing impact of artificial intelligence and machine learning technologies on many facets of life, from commonplace movie recommendations to consequential criminal justice sentencing decisions, has prompted concerns that these systems may behave in an unfair or discriminatory manner [1,2,3]. A number of studies have subsequently demonstrated that bias and fairness issues in AI are both harmful and pervasive [4,5,6]. The AI community has responded by developing a broad array of mathematical formulations of fairness and learning algorithms which aim to satisfy them [7,8,9,10,11,12]. Fairness, however, is not a purely technical construct, having social, political, philosophical, and legal facets [13]. At this juncture, the necessity has become clear for interdisciplinary analyses of fairness in AI and its relationship to society, to civil rights, and to the social goals that mathematical fairness definitions are intended to support, which have not always been made explicit [14].

In particular, it is important to connect fairness and bias in algorithms to the broader context of fairness and bias in society, which has long been the concern of civil rights and feminist scholars and activists [3,15]. In this work, we address the specific challenges of fairness in AI that are motivated by **intersectionality**, an analytical lens from the third-wave feminist movement which emphasizes that civil rights and feminism should be considered simultaneously rather than separately [16]. More concretely, intersectionality analyzes how systems of oppression in our society interact and interlock to create patterns of advantage and disadvantage that affect people differently at the intersection of multiple protected dimensions. Thus, in an AI fairness context, intersectionality implies not only that fairness criteria should consider multiple protected attributes simultaneously, but also that they should account for the impact of these unfair systems of society. While multi-attribute fairness definitions have previously been proposed [17,18], they fall short when it comes to the latter aspect of intersectionality. This research aims to fill this gap.

In this work, we propose **intersectional AI fairness criteria** and perform a *comprehensive, interdisciplinary analysis* of their relation to the concerns of diverse fields including the **humanities**, **law**, **privacy**, **economics**, and **statistical machine learning**. In order to create our quantitative measure of fairness, we must first understand how the principles that underlie intersectionality can be manifested in terms of decision making and in particular how harms are manifested via causal assumptions and how ideal outcomes differ from prior models of fairness. For instance, as we will discuss in Section 3, fairness measures that consider intersections of sensitive attributes or protected classes may not explicitly address how complex and interdependent systems of power and privilege impact those intersecting groups. In Section 2 we will formulate desirable properties that *any* intersectional fairness metric should have, motivating our approach.

We propose a quantitative measure of fairness, called *differential fairness*, and show how it can be used to capture some aspects of intersectionality. In particular, we show how it can be used to quantify harms from fairness violations. To obtain this interpretable guarantee, our approach builds upon ideas from differential privacy, a mathematical quantification of the privacy-preserving properties of an algorithmic mechanism [19,20]. Differential privacy is an information-theoretic privacy notion, in that it is a special case of Rényi differential privacy, a more general notion that is defined in terms of the Rényi divergence [21]. The definition of differential privacy, due to [19,20], follows:

**Definition** **1.**
*M(x) is ϵ-differentially private if*

(1)
$$\frac{P(M(x)\in S)}{P(M\left(x^{\prime }\right)\in S)}\leq e^{\epsilon }$$

*for all outcomes S and pairs of databases x, $x^{\prime }$ differing in a single element.*


Here, M(x) is a randomized algorithmic mechanism, and ϵ is the degree of differential privacy achieved by M(x) (lower is better). Essentially, differential privacy is a promise; if an individual contributes their data to a dataset, their resulting utility, due to algorithms applied to that dataset, will not be substantially affected if ϵ is small. A particular value of ϵ corresponds to an economic guarantee: a bound on the change in the expected utility for any individual who contributes their data. As such, differential privacy *operationalizes algorithmic privacy* in terms of provable guarantees on harms due to the algorithm. Differential privacy thus has *operational significance*, i.e., its measurement represents the construct of “privacy” in a meaningful way as it has a direct relationship to the real-world privacy-related harms that may occur. The potential data contributors and other stakeholders can hence interpret what the value of the differential privacy parameter ϵ means to them.

More generally, mathematical definitions that aim to operationalize abstract constructs ideally should similarly have operational significance. For example, the source coding theorem shows that Shannon entropy operationalizes the notion of uncertainty by providing an interpretation with real-world implications: a bound on the number of bits per symbol that data from a given distribution can be compressed down to. It is natural to ask whether these ideas from algorithmic privacy can be translated to the related problem of algorithmic fairness: *can we **operationalize fairness** with a definition that provides an interpretable guarantee on the fairness harms due to the algorithm?*

In this work, we answer this question in the affirmative. We propose a fairness definition, which we call *differential fairness*, that adapts differential privacy to the fairness setting. Crucially, it provably inherits privacy and economic guarantees that are analogous to those of differential privacy, allowing suitably informed stakeholders to similarly interpret its meaning. Hence, similar to differential privacy, it has *operational significance*, in the sense that it meaningfully represents the construct of “fairness” in terms of bounds on possible harms due to the algorithm. (Here, “possible harms” refers to direct harms that the algorithm produces in an idealized setting. This does not account for additional harms due to its interaction with a larger sociotechnical system [22] or harms that are not captured by the fairness metric itself [23]. See Section 1.3 for further discussion.) The meaningful interpretation of the metric is particularly important for AI fairness, as it is crucial that affected individuals and stakeholders are able to understand their level of risk in order to make informed decisions, both at the personal level and at the policy level. The legal obligation to provide information relating to these risks is trending upwards globally; see, e.g., Europe’s General Data Protection Regulation (GDPR) and the California Consumer Privacy Act (CCPA). (The interpretability properties of the fairness metric do not guarantee that stakeholders with non-technical backgrounds will be able to interpret it correctly. Addressing this is an ongoing research challenge [24,25,26]. See Section 1.3).

### 1.1. Differential Fairness: A Definition

We adapt the notation of [27] to all definitions in this paper. Suppose M(x) is a (possibly randomized) mechanism which takes an instance x∈χ and produces an outcome *y* for the corresponding individual, $S_{1},\ldots ,S_{p}$ are discrete-valued protected attributes, $A=S_{1}\times S_{2}\times \ldots \times S_{p}$, and θ is the distribution which generates x. For example, the mechanism M(x) could be a deep learning model for a lending decision, *A* could be the applicant’s possible *gender* and *race*, and θ the joint distribution of *credit scores* and protected attributes. The protected attributes are included in the attribute vector x, although M(x) is free to disregard them (e.g., if this is disallowed). We suppose that the user of the assigned outcomes (the *vendor*), who may not be the *data owner*, may be *untrusted* and should not access the input data [7]. The setting is illustrated in Figure 1.

Similar to differential privacy, our proposed *differential fairness* definition bounds ratios of probabilities of outcomes resulting from a mechanism. (Differential fairness measures the fairness of the mechanism and not the fairness of a larger sociotechnical system in which it is embedded (see Section 1.3). For example, if the mechanism is an AI that is used to make recommendations to the vendor, it does not account for the vendor injecting additional bias into a subsequent decision. If appropriate for a particular application, human decision making can also be factored into the “mechanism,” thereby measuring the fairness of the human–AI super-system instead of just that of the AI).

**Definition** **2.**
*A mechanism M(x) is ϵ-differentially fair (DF) with respect to (A,Θ) if for all θ∈Θ with x∼θ and y∈Range(M),*

(2)
$$e^{-\epsilon }\leq \frac{P_{M,\theta }\left(M\left(x\right)=y|s_{i},\theta \right)}{P_{M,\theta }\left(M\left(x\right)=y|s_{j},\theta \right)}\leq e^{\epsilon },$$

*for all $(s_{i},s_{j})\in A\times A$ where $P(s_{i}|\theta )>0$, $P(s_{j}|\theta )>0$.*


In Equation (Equation 2), $s_{i}$, $s_{j}\in A$ are tuples of *all* protected attribute values, e.g., *gender*, *race*, and *nationality*, and Θ is a set of distributions θ which could plausibly generate each instance x. (The possibility of multiple θ∈Θ is valuable from a privacy perspective, where Θ is the set of *possible beliefs* that an adversary may have about the data and is motivated by the work of [27]. Continuous protected attributes are also possible, in which case sums are replaced by integrals in our proofs.) For example, Θ could be the set of Gaussian distributions over *credit scores* per value of the protected attributes, with mean and standard deviation in a certain range. The probabilities $P_{M,\theta }$ are over the randomness in the data as well as in the mechanism, so it is possible to achieve the definition when the mechanism is deterministic, unlike for differential privacy.

### 1.2. Our Contribution

While the similarity of the mathematical formulation to differential privacy is clear from a comparison of Equations (Equation 1) and (Equation 2), in what follows we will further motivate this definition from interdisciplinary perspectives, i.e., based on principles from the law and the humanities. In particular, we will provide arguments that motivate differential fairness as upholding the values associated with *intersectionality* [16], a framework for examining fairness, which we will discuss in Section 2.

Our contributions include:A critical analysis of the consequences of intersectionality in the particular context of fairness for AI (Section 2);Three novel fairness metrics: *differential fairness (DF)*, which aims to uphold intersectional fairness for AI and machine learning systems (Section 4), *DF bias amplification*, a slightly more conservative fairness definition than DF (Section 6), and *differential fairness with confounders (DFC)* (Section 7);Illustrative worked examples to aid with understanding (Section 5);Proofs of the desirable intersectionality, privacy, economic, and generalization properties of our metrics (Section 8);A simple learning algorithm which enforces our criteria using batch or deterministic gradient methods (Section 9.1);A practical and scalable learning algorithm that enforces our fairness criteria using stochastic gradient methods (Section 9.2);Extensive experiments on census, criminal recidivism, hospitalizations, and loan application data to demonstrate our methods’ practicality and benefits (Section 10).

As AI fairness is a problem that extends well beyond the boundaries of traditional computer science, our work aims to cross disciplinary boundaries in order to tackle it. We view the significance of this work as more than simply the introduction of mathematical definitions and associated algorithms and proofs. Our definitions and proofs build upon existing work on differential privacy, which leads to results that are elegant in their simplicity and yet are relatively straightforward. We do not claim that the technical novelty of our results or the difficulty of proofs are the most significant aspects of the research. Rather, we view the primary significance of our proposed differential fairness definition as threefold:It operationalizes AI fairness in a manner that is more meaningful than previous definitions, in the sense that it can be interpreted in terms of economic and privacy guarantees, leveraging connections to differential privacy.It is the first AI fairness definition to implement the principles and values behind intersectionality, an important notion of fairness from the humanities and legal literature. As such, part of the significance of this work is philosophical, not only mathematical.It is a stepping stone toward the holistic, trans-disciplinary research endeavor that is necessary to adequately solve the real-world sociotechnical problem of AI fairness.

### 1.3. Limitations and Caveats

In order to properly contextualize our contributions, we must consider the limitations of the use of AI fairness metrics, including but not limited to our proposed metrics, in pursuit of the real-world social goal of *fairness*, or more specifically, *intersectional fairness*. In general terms, because AI algorithms are embedded in complicated sociotechnical systems, enforcing a fairness definition—even one with provable guarantees—is not generally sufficient to ensure that the overall decision-making process, and its higher-order consequences, are necessarily fair. We discuss several aspects of these limitations and challenges below.

**Accounting for the Sociotechnical Context:** Algorithmic systems do not generally exist in a vaccuum. They are deployed within the context of larger real-world sociotechnical systems. Our provable guarantees on the harms caused by an algorithmic system pertain to the direct impact of the system’s output, but not to the complicated interaction between that algorithm and its broader context, which can be difficult to account for [22].

One important aspect of the sociotechnical context is whether the AI algorithm’s output is the final decision that is made on an individual or whether it is used to support such a decision that is ultimately made by a human or another algorithm. For example, in AI-generated criminal justice risk assessments such as those made by the COMPAS system [4], the risk score for an incarcerated individual is typically given to a judge who takes it into account, along with additional information on the individual and their case, in order to make bail and sentencing decisions.

Fairness issues can arise at the nexus between a human decision maker and the AI. For example, Green and Chen [28] found that the presence of an algorithmic risk assessment leads humans to increase the prioritization of risk at the expense of other factors. Humans are frequently poor at overseeing AI systems due to issues such as *automation bias* (deferring to an algorithm even when it is not appropriate), and human oversight can legitimize flawed human–AI decision-making processes [29]. Like other AI fairness metrics, differential fairness does not account for such issues.

Indeed, our theoretical analysis assumes that *the decisions made on the individuals correspond to the output of the mechanism, and not decisions made by a human decision maker who is merely informed by the mechanism*. More specifically, we assume a particular protocol in which the individuals’ data are kept on a secure server and the output of the mechanism is the only information available to the user of these outputs, referred to as the *vendor* (see Figure 1). If this protocol is not respected, e.g., if the vendor is allowed to look at the individuals’ data and could choose to completely ignore the output of the mechanism in making decisions, or deviate from the algorithm’s outputs in systematically biased ways, all bets are clearly off regarding the fairness of the overall sociotechnical system [22]. In the scenario where a judge uses knowledge of the defendant in addition to the AI-generated risk score to make a bail or sentencing decision, our assumed protocol has been violated (since the vendor, i.e., the judge, looked at the individual’s data, and they made a decision that differed from the algorithm’s output), and so the overall sociotechnical decision-making process will not enjoy the provable guarantees afforded by the differential fairness definition, as measured on the AI risk scoring system.

One way around this is to view the human decision maker as part of the mechanism. We can straightforwardly estimate the differential fairness of this human–AI super-system empirically (see Section 4.2). However, our learning algorithm would not be applicable since it is not currently possible to train a human via backpropagation!

**Mismatch Between Fairness Metrics and Real-World Fairness Harms:** The measurement of fairness depends heavily on assumptions made when modeling the problem at hand, particularly when defining and measuring class labels that encode abstract constructs and are intended to act as proxies for decisions. For example, the COMPAS system defines its prediction target, *recidivism*, as a new misdemeanor or felony arrest within two years [4]. This assumption conflates arrests with crimes, thereby encoding systemic bias in policing into the class label itself [23]. Due to these types of issues, there is generally a gap in alignment between mathematical fairness metrics and the desired fairness construct regarding the real-world harms that they are intended to measure and prevent [23]. Whenever we interpret our mathematical guarantees on fairness-related harms due to an algorithm, it is important to recognize that the mathematical definition may not perfectly align with real-world harms.**Countering Structural Oppression:** Recently, Kong [30] critiqued fairness definitions based on multiple tuples of attributes [17,18,31] regarding the extent to which they implement intersectionality. It was argued that parity-based fairness does not place the proper emphasis on structural oppression, which is a core tenet of intersectionality, and may not go far enough to rectify such oppression. We broadly agree with this statement. However, as we argue here and in prior work [32], parity-based fairness is an important starting point toward addressing structural oppression which is appropriate for many applications. In particular, we try to explicitly incorporate the causal structure of interlocking structural oppression. We believe our definition can be extended to correct societal unfairness beyond the point of parity, as advocated by [30]. However, we leave this for future work.**Stakeholder Interpretation:** We have motivated our work via the interpretation that our fairness definition’s provable guarantees provide. We note, however, that it is not always straightforward to ensure that stakeholders correctly interpret the fairness properties of AI systems, particularly when they do not have technical backgrounds, and there is a growing body of research on this challenge and how best to address it [24,25,26]. Studying how stakeholders interpret and perceive differential fairness is beyond the scope of this paper, but is an important question for future research.**Contextual Considerations:** The impact of systems of oppression and privilege may vary contextually. For example, gender discrimination varies widely by area of study and this is a major factor in the choices women make in selecting college majors [33]. Similarly, in some contexts individuals from a particular race, gender, or sexual orientation may be able to “pass” as another, thus potentially avoiding discrimination, and in other contexts where their “passing” status is revealed, those individuals may suffer additional discrimination [34]. Our fairness framework does not presently consider these contextual factors.**Knowledge of Protected Attributes:** In order to compute (or enforce) our differential fairness criterion, we must observe the individuals’ protected attributes. This is not always possible in certain applications. For instance, social media users may not declare their gender, race, or other protected demographic information. Future work could address the extension of our methods to handle this scenario, e.g., via an imputation strategy.

## 2. Intersectionality and Fairness in AI

Our definition(s) differ *quantitatively* from prior fairness definitions (which we address in Section 3) but more importantly represent a *qualitatively* different approach to defining fairness. In order to understand what an intersectional analysis would ask from a fairness criterion, we must describe the challenges posed by legal, political, and humanistic analyses of fairness and discrimination. We now turn to describing these challenges and our desiderata for any fairness definition which seeks to address them.

Intersectionality is a lens for examining societal unfairness which originally arose from the observation that sexism and racism have intertwined effects, in that the harm done to Black women by these two phenomena is more than the sum of the parts [16,35]. The notion of intersectionality was later extended to include overlapping injustices along more general axes [36]. In its general form, intersectionality emphasizes that systems of oppression built into society lead to *systematic disadvantages along intersecting dimensions*, which include not only gender, but also race, nationality, sexual orientation, disability status, and socioeconomic class [16,35,36,37,38,39]. (Please note that bell hooks prefers that her name be written in lower case, cf. https://www.nytimes.com/2006/09/28/books/28chic.html accessed on 4 April 2023.) These systems are interlocking in their effects on individuals at *each intersection of the affected dimensions*.

The term *intersectionality* was introduced by Kimberlé Crenshaw in the 1980s [16] and popularized in the 1990s, e.g., by Patricia Hill Collins [36], although the ideas are much older [35,37]. In the context of machine learning and fairness, intersectionality was considered by Buolamwini and Gebru [5], who studied the impact of the intersection of gender and skin color on computer vision performance, by Kearns et al. [17] and Hebert-Johnson et al. [18], who aimed to protect certain subgroups in order to prevent “fairness gerrymandering,” and by Yang et al. [40], who considered algorithmic fairness across multiple overlapping groups simultaneously. From a humanities perspective, Noble [3] critiqued the behavior of Google search with an intersectional lens by examining the search results for terms relating to women, people of color, and their intersections, e.g.,  “Black girls.” Very recently La Cava et al. [41] proposed intersectional fairness definitions combining our differential fairness work and multicalibration [18], and Lett and La Cava [42] advanced a perspective on how intersectionality informs AI fairness in healthcare applications.

Intersectionality has implications for AI fairness beyond the use of multiple *protected attributes*. Many fairness definitions aim (implicitly or otherwise) to uphold the principle of **infra-marginality**, which states that differences between protected groups in the distributions of “merit” or “risk” (e.g., the probability of carrying contraband at a police stop) should be taken into account when determining whether bias has occurred [43]. (More specifically, Simoiu et al. [43] argue that risk scores are meritocratic, that fairness corresponds to thresholding risk scores at the same point for different demographics, and that parity-based notions of fairness are hence problematic because they may be incompatible with this version of fairness. They refer to this concern as the *infra-marginality problem*. While “infra-marginality” refers to a technical point, we use the term to also refer to the philosophical viewpoint underlying this argument which emphasizes meritocracy and takes an anti-parity stance. For more discussion, please see [32].) A closely related argument is that parity of outcomes between groups is at odds with accuracy [7,8]. Intersectionality theory provides a counterpoint: these differences in risk/merit, while acknowledged, are frequently due to systemic structural disadvantages such as racism, sexism, inter-generational poverty, the school-to-prison pipeline, mass incarceration, and the prison-industrial complex [16,37,38,44,45]. Systems of oppression can lead individuals to perform below their potential, for instance, by reducing available cognitive bandwidth [46] or by increasing the probability of incarceration [44,47]. In short, the *infra-marginality* principle makes the implicit assumption that society is a fair, level playing field, and thus differences in “merit” or “risk” between groups in data and predictive algorithms are often to be considered legitimate. In contrast, *intersectionality* theory posits that these **distributions of merit and risk are often influenced by unfair societal processes** (see Figure 2).

As an example of a scenario affected by unfair processes, consider the task of predicting prospective students’ academic performance for use in college admissions decisions. As discussed in detail by [46] and references therein, individuals belonging to marginalized and non-majority groups are disproportionately impacted by challenges of poverty and racism (in its structural, overt, and covert forms), including chronic stress, access to healthcare, under-treatment of mental illness, micro-aggressions, stereotype threat, disidentification with academics, and belongingness uncertainty. Similarly, LGBT and especially transgender, non-binary, and gender non-conforming students disproportionately suffer bullying, discrimination, self-harm, and the burden of concealing their identities. These challenges are often further magnified at the intersection of affected groups. A survey of 6,450 transgender and gender non-conforming individuals found that the most serious discrimination was experienced by people of color, especially Black respondents [48]. Verschelden explains the impact of these challenges as a tax on the “cognitive bandwidth” of minoritized students, which in turn affects their academic performance. She states that the evidence is clear


*“…that racism (and classism, homophobia, etc.) has made people physically, mentally, and spiritually ill and dampened their chance at a fair shot at higher education (and at life and living).”*


A classifier trained to predict students’ academic performance from historical data hence aims to emulate outcomes that were substantially affected by unfair factors [1]. An *accurate predictor* for a student’s GPA may therefore not correspond to a *fair decision-making procedure* [49]. We can resolve this apparent conflict if we are careful to distinguish between the *statistical problem* of classification and the *economic problem* of the assignment of outcomes (e.g., admission decisions) to individuals based on classification. (Recall that we assume the outputs of the mechanism correspond to the final decisions made on the individuals.) Viewing the classifier’s task as a policy question, it becomes clear that high accuracy need not be the primary goal of the system, especially when we consider that “accuracy” is measured on unfair data. (Amazon abandoned a job candidate classifier which was found to be gender-biased [50]. We speculate that this was likely due to similar issues).

In Figure 2 we summarize the causal assumptions regarding society and data and the idealized “perfect world” scenarios implicit in the two approaches to fairness. Infra-marginality *(a)* emphasizes that the distribution over relevant attributes *X* varies across protected groups *A*, which leads to potential differences in so-called “merit” or “risk” between groups, typically presumed to correspond to latent ability and thus “deservedness” of outcomes *Y* [43]. Intersectionality *(b)* emphasizes that we must also account for systems of oppression which lead to (dis)advantage at the intersection of multiple protected groups, impacting all aspects of the system including the ability of individuals to succeed (“merit”) to their potential, had they not been impacted by (dis)advantage [16]. In the ideal world that an algorithmic (or other) intervention aims to achieve, infra-marginality-based fairness desires that individual “merit” be the sole determiner of outcomes *(c)* [8,43], which can lead to disparity between groups [7]. In ideal intersectional fairness *(d)*, since ability to succeed is affected by unfair processes, it is desired that this unfairness be corrected and individuals achieve their true potential [46]. Assuming potential does not substantially differ across protected groups, this implies that parity between groups is typically desirable. (Disparity could still be desirable if there are legitimate confounders which depend on protected groups, e.g., choice of department that individuals apply to in college admissions. We address this scenario in Section 7).

In light of the above, we argue that an *intersectional* definition of fairness in AI should satisfy the following criteria:Multiple protected attributes should be considered.**All** of the intersecting values of the protected attributes, e.g., *Black women*, should be protected by the definition.The definition should still also ensure that protection is provided on individual protected attribute values, e.g., *women*.The definition should ensure protection for minority groups, who are particularly affected by discrimination in society [36].The definition should ensure that systematic differences between the protected groups, assumed to be due to structural oppression, are rectified, rather than codified.These desiderata do not uniquely specify a fairness definition, but they provide a set of guidelines to which legal, political, and contextual considerations can then be applied to determine an appropriate fairness measure for a particular task.

## 3. Related Work on Fairness

Before providing our new fairness definition, we discuss some of the existing fairness definitions and their relation to the aforementioned criteria. An overview of AI fairness research can be found in [49].

### 3.1. Models for Fairness

**The 80% rule:** Our criterion is related to the *80% rule*, also known as the *four-fifths rule*, a guideline for identifying unintentional discrimination in a legal setting which identifies disparate impact in cases where $P\left(y=1|s_{1}\right)/P\left(y=1|s_{2}\right)\leq 0.8$, for a favorable outcome y=1, disadvantaged group $s_{1}$, and best performing group $s_{2}$ [51]. This corresponds to testing that ϵ≥−log0.8=0.2231, in a version of Equation (Equation 2) where only the outcome y=1 is considered.**Demographic Parity:** Dwork et al. [7] defined (and criticized) the fairness notion of *demographic parity*, also known as *statistical parity*, which requires that $P\left(y|s_{i}\right)=P\left(y|s_{j}\right)$ for any outcome *y* and pairs of protected attribute values $s_{i}$, $s_{j}$ (here assumed to be a single attribute). This can be relaxed, e.g., by requiring the total variation distance between the distributions to be less than ϵ. Differential fairness is closely related, as it also aims to match probabilities of outcomes but measures differences using ratios and allows for multiple protected attributes. The criticisms of [7] are mainly related to ways in which subgroups of the protected groups can be treated differently while maintaining demographic parity, which they call “*subset targeting*,” and which [17] term “*fairness gerrymandering*”. Differential fairness explicitly protects the intersection of multiple protected attributes, which can be used to mitigate some of these abuses.**Equalized Odds:** To address some of the limitations with demographic parity, Hardt et al. [8] propose to instead ensure that a classifier has equal error rates for each protected group. This fairness definition, called *equalized odds*, can loosely be understood as a notion of “demographic parity for error rates instead of outcomes.” Unlike demographic parity, equalized odds rewards accurate classification, and penalizes systems only performing well on the majority group. However, theoretical work has shown that equalized odds is typically incompatible with correctly calibrated probability estimates [52]. It is also a relatively weak notion of fairness from a civil rights perspective compared to demographic parity, as it does not ensure that outcomes are distributed equitably. Hardt et al. also propose a variant definition called *equality of opportunity*, which relaxes equalized odds to only apply to a “deserving” outcome. It is straightforward to extend differential fairness to a definition analogous to equalized odds, although we leave the exploration of this for future work. A more recent algorithm for enforcing equalized odds and equality of opportunity for kernel methods was proposed by [53].**Individual Fairness (“Fairness Through Awareness”):** The *individual fairness* definition, due to [7], mathematically enforces the principle that *similar individuals should achieve similar outcomes* under a classification algorithm. An advantage of this approach is that it preserves the privacy of the individuals, which can be important when the user of the classifications (the *vendor*), e.g., a banking corporation, cannot be trusted to act in a fair manner. However, this is difficult to implement in practice as one must define “similar” in a fair way. The individual fairness property also does not necessarily generalize beyond the training set. In this work, we take inspiration from Dwork et al.’s *untrusted vendor* scenario and the use of a privacy-preserving fairness definition to address it.**Counterfactual Fairness:** Kusner et al. [54] propose a causal definition of fairness. Under their *counterfactual fairness* definition, changing protected attributes *A*, while holding things which are not causally dependent on *A* constant, will not change the predicted distribution of outcomes. While theoretically appealing, there are difficulties in implementing this in practice. First, it requires an accurate causal model at the fine-grained individual level, while even obtaining a correct population-level causal model is generally very difficult. To implement it, we must solve a challenging causal inference problem over unobserved variables, which generally requires approximate inference algorithms. Finally, to achieve counterfactual fairness, the predictions (usually) cannot make direct use of any descendant of *A* in the causal model. This generally precludes using *any of the observed features* as inputs.**Threshold Tests:** Simoiu et al. [43] address *infra-marginality* by modeling risk probabilities for different subsets (i.e., attribute values) within each protected category and requiring algorithms to threshold these probabilities at the same points when determining outcomes. In contrast, based on *intersectionality* theory, our proposed differential fairness criterion specifies protected categories whose intersecting subsets should be treated equally, regardless of differences in risk across the subsets. Our definition is appropriate when the differences in risk are due to structural systems of oppression, i.e., the risk probabilities themselves are impacted by an unfair process. We also provide a *bias amplification* version of our metric, following [10], which is more in line with the infra-marginality perspective, namely that differences in probabilities of outcomes for different groups may be considered legitimate.

### 3.2. Subgroup Fairness and Multicalibration

Relevant fairness definitions aim to detect and prevent discriminatory bias with respect to a *set of protected attributes*, such as *gender*, *race*, and *disability status*. Given criterion A, we focus on multi-attribute definitions. The two dominant multi-attribute approaches in the literature are *subgroup fairness* [17] and *multicalibration* [18].

**Definition** **3.**(Statistical Parity Subgroup Fairness [17]). *Let G be a collection of protected group indicators g:A→{0,1}, where g(s)=1 designates that an individual with protected attributes s is in group g. Assume that the classification mechanism M(x) is binary, i.e., y∈{0,1}.*
*Then M(x) is γ-statistical parity subgroup fair with respect to θ and G if for every g∈G,*

(3)
$$|P_{M,\theta }\left(M\left(x\right)=1\right)-P_{M,\theta }\left(M\left(x\right)=1|g\left(s\right)=1\right)|\times P_{\theta }\left(g\left(s\right)=1\right)\leq \gamma .$$



Note that for γ∈[0,1], smaller is better. The first term penalizes a difference between the probability of the *positive* class label for group *g* and the population average of this probability. The term $P_{\theta }\left(g\left(s\right)=1\right)$ weights the penalty by the size of group *g* as a proportion of the population. *Statistical parity subgroup fairness* (*SF*) is a multi-attribute definition satisfying criterion A. To satisfy B and C criteria, G can be *all* intersectional subgroups (e.g., *Black women*) and top-level groups (e.g., *men*). The first term in Equation (Equation 3), which encourages similar outcomes between groups, enforces criterion E.

From an intersectional perspective, one concern with SF is that it does not satisfy criterion D, the protection of minority groups. The term $P_{\theta }\left(g\left(s\right)=1\right)$ weights the “per-group (un)fairness” for each group *g*, i.e., Equation (Equation 3) applied to *g* alone, by its proportion of the population, thereby *specifically downweighting the consideration of minorities*. In Figure 3, we show an example where varying the size of a minority group $P_{\theta }\left(g\left(minority\right)=1\right)$ drastically alters γ-subgroup fairness, which finds that **a rather extreme scenario is more acceptable when the minority group is small**. Our proposed criterion, ϵ-DF (introduced in Section 4), is constant in $P_{\theta }\left(g\left(minority\right)=1\right)$.

Figure 4 reports “per-group” γ’s on the UCI Adult census dataset, i.e., Equation (Equation 3) applied separately to each group is empirically seen to have an increasing relationship with P(group) (simplified notation of $P_{\theta }\left(g\left(minority\right)=1\right)$). The final γ-SF is determined by the worst case of the per-group γ’s. *A small minority group thereby will most likely not directly affect*γ-SF, since the downweighting makes it unlikely to be the “most unfair” group.

Kearns et al. [17] justify the use of the $P_{\theta }\left(g\left(s\right)=1\right)$ term via statistical considerations, as it is useful to prove generalization guarantees to extrapolate from empirical estimates of γ (see Section 8.4). From a different ethical perspective, total utilitarianism, increasing the utility (i.e., reducing unfairness) for a large group of individuals at the expense of smaller groups (i.e., downweighting small minority groups) could also be justified by the increase in the total utility of the population. The problem with total utilitarianism, of course, is that it admits a scenario where many people possess low utility (i.e., higher unfairness). We do not intend to dismiss SF as a valid notion of fairness. Our claim here, rather, is simply that due to its treatment of minority groups, SF does not fully encapsulate the principles of fairness advocated by intersectional feminist scholars and activists [16,35,36,38,39].

Other candidate multi-attribute fairness definitions include *false positive subgroup fairness* [17] and *multicalibration* [18]. These definitions are similar to SF, but they concern false positive rates and calibration of prediction probabilities, respectively. Since they focus on reliability of estimation rather than allocation of outcomes, they do not directly address the issues (criterion E) raised by the civil rights/feminist perspective. This does not preclude their use for intersectional fairness scenarios in which harms are caused by incorrect predictions (e.g., higher error rates for unprivileged groups), rather than unfair outcome assignments (e.g., higher false positive rates for privileged groups); indeed, this is the type of approach [5] take for studying intersectional fairness in computer vision applications. Nevertheless, we will not consider them further here.

## 4. Differential Fairness (DF)

We now introduce our proposed fairness measures which satisfy our intersectionality criteria. There are multiple conceivable fairness definitions which satisfy these criteria. For example, SF could be adapted to address criterion D by simply dropping the $P_{\theta }\left(g\left(s\right)=1\right)$ term, at the loss of its associated generalization guarantees. We instead select an alternative formulation, which is similar to this approach in spirit, but which has additional beneficial properties from a societal perspective regarding the *law*, *privacy*, and *economics*, as we shall discuss below. The privacy and economic guarantees enjoyed by our definition demonstrate that it has operational significance, in that it provably prevents harms, analogously to differential privacy. (Not withstanding the gap between the behavior of an idealized algorithmic system and the complicated impacts of deploying it within a broader real-world sociotechnical system [22] and the gap between fairness metrics and the real-world harms they are intended to measure [23]. See Section 1.3.) Our formalism has a particularly elegant *intersectionality property*, in that Criterion C (protecting higher-level groups) follows automatically from Criterion B (protecting intersectional subgroups).

We motivate our criteria from a legal perspective. Consider the 80% rule, established in the Code of Federal Regulations [51] as a guideline for establishing disparate impact in violation of anti-discrimination laws such as Title VII of the Civil Rights Act of 1964. The 80% rule states that there is legal evidence of adverse impact if the ratio of probabilities of a particular favorable outcome, taken between a disadvantaged and an advantaged group, is less than 0.8:(4)$$\begin{array}{c} P(M(x)=1|groupA)/P(M(x)=1|groupB)<0.8. \end{array}$$

Our first proposed criterion, which we call **differential fairness (DF)**, extends the 80% rule to protect multi-dimensional intersectional categories, with respect to multiple output values. We similarly restrict ratios of outcome probabilities between groups, but instead of using a predetermined fairness threshold at 80%, we measure fairness on a sliding scale that can be interpreted similarly to that of *differential privacy*, a definition of privacy for data-driven algorithms [19]. Differential fairness measures the **fairness cost** of mechanism M(x) with a parameter ϵ. Combining these elements, we obtain our differential fairness criterion, which we stated above as Definition 2.

This is an intuitive **intersectional definition of fairness**: *regardless of the combination of protected attributes, the probabilities of the outcomes will be similar*, as measured by the ratios versus other possible values of those variables, for small values of ϵ. For example, the probability of being given a loan would be similar regardless of a protected group’s intersecting combination of *gender*, *race*, and *nationality*, marginalizing over the remaining attributes in x. If the probabilities are always equal, then ϵ=0; otherwise, ϵ>0.

### 4.1. Relationship to Privacy Definitions

**Differential Privacy:** Like differential privacy [19,20], the differential fairness definition bounds ratios of probabilities of outcomes resulting from a mechanism. There are several important differences, however. When bounding these ratios, differential fairness considers different values of a set of protected attributes, rather than databases that differ in a single element. It posits a specified set of possible distributions which may generate the data, while differential privacy implicitly assumes that the data are independent [27]. Finally, since differential fairness considers randomness in data as well as in the mechanism, it can be satisfied with a deterministic mechanism, while differential privacy can only be satisfied with a randomized mechanism.

The study in Jagielski et al. [55] developed methods which aim to satisfy differential privacy and equalized-odds fairness simultaneously. This is a different goal to our work, which aims to ensure intersectional fairness using a fairness definition that was designed leveraging differential privacy.

**Pufferfish:** We have arrived at our criterion based on the 80% rule, but it can also be derived as an application of *pufferfish* [27], a generalization of differential privacy [20] which uses a variation of Equation (Equation 2) to hide the values of an arbitrary set of secrets. Different choices of the secrets and the data that the Pufferfish privacy-preserving mechanism operates on lead to both differential privacy and differential fairness.

**Definition** **4.**
*A mechanism M(x) is ϵ-pufferfish private [27] in a framework (S,Q,Θ) if for all θ∈Θ with x∼θ, for all secret pairs $(s_{i},s_{j})\in Q$ and y∈Range(M),*

(5)
$$e^{-\epsilon }\leq \frac{P_{M,\theta }\left(M\left(x\right)=y|s_{i},\theta \right)}{P_{M,\theta }\left(M\left(x\right)=y|s_{j},\theta \right)}\leq e^{\epsilon },$$

*when $s_{i}$ and $s_{j}$ are such that $P(s_{i}|\theta )>0$, $P(s_{j}|\theta )>0$.*


While the formulation for differential fairness is mathematically very similar to Pufferfish, it is used with the goal of obtaining fairness, which leads to its more specific instantiation of the Pufferfish secrets (protected attributes), the input data (each individual’s feature vectors), and the mechanism (the behavior of the classifier itself under the distribution of the data).

### 4.2. Empirical DF Estimation

A major challenge for measuring fairness in an intersectional context, either via ϵ-DF (differential fairness), γ-SF (subgroup fairness), or related notions, is to estimate M(x)’s marginal behavior $P_{M,\theta }\left(y|s,\theta \right)$ for each (y,s) pair, with potentially little data for each of these [56]. If $P_{M,\theta }$ is unknown, it can be estimated using the empirical distribution, or via a probabilistic model of the data. Assuming discrete outcomes, $P_{Data}\left(y|s\right)=\frac{N_{y,s}}{N_{s}}$, where $N_{y,s}$ and $N_{s}$ are empirical counts of their subscripted values in the dataset *D*. **Empirical differential fairness (EDF)** corresponds to verifying that for any *y*, $s_{i}$, $s_{j}$, we have
(6)$$e^{-\epsilon }\leq \frac{N_{y,s_{i}}}{N_{s_{i}}}\frac{N_{s_{j}}}{N_{y,s_{j}}}\leq e^{\epsilon },$$
whenever $N_{s_{i}}>0$ and $N_{s_{j}}>0$. However, in the intersectional setting, the counts $N_{y,s}$ at the intersection of the values of the protected attributes become rapidly smaller as the dimensionality and cardinality of protected attributes increase. The $N_{y,s}$ counts may even be 0, which can make the estimate of ϵ in Equation (Equation 6) infinite/undefined.

Alternatively, to address the zero count issue, we can estimate ϵ-DF via the posterior predictive distribution of a Dirichlet-multinomial as
(7)$$e^{-\epsilon }\leq \frac{N_{y,s_{i}}+\alpha }{N_{s_{i}}+\left|Y\right|\alpha }\frac{N_{s_{j}}+\left|Y\right|\alpha }{N_{y,s_{j}}+\alpha }\leq e^{\epsilon },$$
where scalar α is each entry of the parameter of a symmetric Dirichlet prior with concentration parameter |Y|α, Y=Range(M). We refer to this as **smoothed EDF**.

Note that EDF and smoothed EDF methods can sometimes be unstable in extreme cases when nearly all instances are assigned to the same class. To address this issue, instead of using empirical hard counts per group $N_{y,s}$, we can also use *soft counts* for (smoothed) EDF, based on a probabilistic classifier’s predicted P(y|x), as follows:(8)$$e^{-\epsilon }\leq \frac{\sum_{x\in D:A=s_{i}}P\left(y|x\right)+\alpha }{N_{s_{i}}+\left|Y\right|\alpha }\frac{N_{s_{j}}+\left|Y\right|\alpha }{\sum_{x\in D:A=s_{j}}P\left(y|x\right)+\alpha }\leq e^{\epsilon }.$$

## 5. Illustrative Worked Examples

A simple worked example of differential fairness is given in Figure 5. In the example, given an applicant’s score *x* on a standardized test, the mechanism M(x)=x≥t approves the hiring of a job applicant if their test score x≥t, with t=10.5. The scores are distributed according to θ, which corresponds to the following process. The applicant’s protected group is 1 or 2 with probability 0.5. Test scores for group 1 are normally distributed $N(x;\mu_{1}=10,\sigma =1)$, and for group 2 they are distributed $N(x;\mu_{2}=12,\sigma =1)$. In the figure, the group-conditional densities are plotted on the top, along with the threshold for the hiring outcome being *yes* (i.e., M(x)=1). Shaded areas indicate the probability of a *yes* hiring decision for each group (overlap in purple). On the bottom, the calculations show that M(x) is ϵ-differentially fair for ϵ=2.337. This means that the probability ratios are bounded within the range $\left(e^{-\epsilon },e^{\epsilon }\right)=\left(0.0966,10.35\right)$, i.e., one group has around 10 times the probability of some particular hiring outcome than the other (y= *no*). Presuming that the two groups are roughly equally capable of performing the job overall, this is clearly unsatisfactory in terms of fairness.

Differential fairness specifically addresses the *intersectional* setting by considering outcome probabilities at each intersection of a set of protected variables. We illustrate this with an example on admissions of prospective students to a particular University X (Table 1). The protected attributes are *gender* and *race*, and the mechanism is the admissions process, with a binary outcome. Our data are adapted from a real-world scenario involving treatments for kidney stones, often used to demonstrate Simpson’s paradox [57,58]. Here, the “paradox” is that for *race 1*, individuals of *gender A* are more likely to be admitted than those of *gender B*, and for *race 2*, those of *gender A* are also more likely to be admitted than those of *gender B*, yet counter-intuitively, *gender B* is more likely to be admitted overall.

Since the admissions process is a black box, we model it using Equation (Equation 6), empirical differential fairness (EDF). By calculating the log probability ratios of (gender,race) pairs from Table 1, as well as for the pairs of probabilities for the *declined admission* outcome (1−P(admit)), and plugging them into Equation (Equation 6), we see that the mechanism is ϵ=1.511-DF with A=gender×race. By calculating ϵ using the admission probabilities in the *overall* row (gender) and the *overall* column (race), we find that ϵ=0.2329 for A=gender, and ϵ=0.8667 for A=race. We will prove in Theorem 2 that ϵ with A=gender×race is an upper bound on ϵ-DF for A=gender and for A=race. Thus, even with a “Simpson’s reversal,” differential (un)fairness will not increase after summing out a protected attribute.

## 6. DF Bias Amplification Measure

We can adapt DF to measure fairness in data, i.e., outcomes assigned by a black box algorithm or social process, by using (a model of) the data’s generative process as the mechanism.

**Definition** **5.**
*A labeled dataset $D=\{\left(x_{1},y_{1}\right),\ldots ,\left(x_{N},y_{N}\right)\}$ is ϵ-differentially fair (DF) in A with respect to model $P_{Model}\left(x,y\right)$ if mechanism $M\left(x\right)=y∼P_{Model}\left(y|x\right)$ is ϵ-differentially fair with respect to $\left(A,\left\{P_{Model}\left(x\right)\right\}\right)$, for $P_{Model}$ trained on the dataset.*


Similar to differential privacy, differences $\epsilon_{2}-\epsilon_{1}$ between two mechanisms $M_{2}\left(x\right)$ and $M_{1}\left(x\right)$ are meaningful (for fixed *A* and Θ and for tightly computed minimum values of ϵ) and measure the additional “fairness cost” of using one mechanism instead of the other. When $\epsilon_{1}$ is the differential fairness of a labeled dataset and $\epsilon_{2}$ is the differential fairness of a classifier measured on the same dataset, $\epsilon_{2}-\epsilon_{1}$ is a measure of the extent to which the classifier increases the unfairness over the original data, a phenomenon that [10] refer to as *bias amplification*.

**Definition** **6.**
*A mechanism M(x) satisfies $\left(\epsilon_{2}-\epsilon_{1}\right)$-DF bias amplification with respect to (A,Θ,D,M) if it is $\epsilon_{2}$-DF and labeled dataset D is $\epsilon_{1}$-DF in A with respect to model M.*


Politically speaking, ϵ-DF is a relatively progressive notion of fairness which we have motivated based on intersectionality (*disparities in societal outcomes are largely due to systems of oppression*) and which is reminiscent of demographic parity [7]. On the other hand, $\left(\epsilon_{2}-\epsilon_{1}\right)$-DF
**bias amplification** is a more **conservative fairness metric** than DF which does not seek to correct unfairness in the original dataset (i.e., it relaxes criterion E), in line with the principle of **infra-marginality** (*a system is biased only if disparities in its behavior are worse than those in society, which are presumed to be legitimate*) [43]. Informally, $\epsilon_{2}$-DF and $\left(\epsilon_{2}-\epsilon_{1}\right)$-DF bias amplification represent “upper and lower bounds” on the unfairness of the system in the case where the relative effect of structural oppression on outcomes is unknown.

## 7. Dealing with Confounder Variables

As we have seen, ϵ-DF measures inequity between protected groups and their intersections at different levels of granularity, although it does not determine whether the inequities were due to systemic factors and/or discrimination. *Confounding variables* can potentially be a legitimate source of disparity. For example, a study on U.C. Berkeley admissions in the 1970s [59] found that a disparity in overall admission rates between men and women was due to a confounding variable, the department that a prospective student chooses to apply to, with women being more likely to apply to more selective departments. This scenario was similar to the worked example in Table 1. With confounders, parity in outcomes between intersectional protected groups, which ϵ-DF rewards, may no longer be desirable (see Figure 6). We propose an alternative fairness definition for when known confounders are present.

**Definition** **7.**
*Let θ∈Θ be distributions over (x,c), where c∈C are confounder variables. A mechanism M(x) is ϵ-differentially fair with confounders (DFC) with respect to (A,Θ,C), if for all c∈C, M(x) is ϵ-DF with respect to $\left(A,\Theta_{|c}\right)$, where $\Theta_{|c}=\left\{P\left(x|\theta ,c\right)|\theta \in \Theta \right\}$.*


In the university admissions case, Definition 7 penalizes disparity in admissions at the department level, and the most unfair department determines the overall unfairness ϵ-DFC.

**Theorem** **1.**
*Let M be an ϵ-DFC mechanism in (A,Θ,C), then M is ϵ-differentially fair in (A,Θ).*


**Proof.** Let θ∈Θ, y∈Range(M), c∈C, and $(s_{i},s_{j})\in A\times A$ where $P(s_{i}|\theta )>0$ and $P(s_{j}|\theta )>0$. We have:
(9)$$\begin{array}{cc} \; & \frac{P_{M,\theta }\left(M\left(x\right)=y|s_{i},\theta \right)}{P_{M,\theta }\left(M\left(x\right)=y|s_{j},\theta \right)}\\ \; & =\frac{\sum_{c\in C}P_{M,\theta }\left(M\left(x\right)=y|s_{i},c,\theta \right)P_{M,\theta }\left(c|s_{i},\theta \right)}{\sum_{c\in C}P_{M,\theta }\left(M\left(x\right)=y|s_{j},c,\theta \right)P_{M,\theta }\left(c|s_{j},\theta \right)}\\ \; & =\frac{\sum_{c\in C}\frac{P_{M,\theta }\left(M\left(x\right)=y|s_{i},c,\theta \right)}{P_{M,\theta }\left(M\left(x\right)=y|s_{j},c,\theta \right)}P_{\theta }\left(c|s_{i},\theta \right)}{\sum_{c\in C}\frac{P_{M,\theta }\left(M\left(x\right)=y|s_{j},c,\theta \right)}{P_{M,\theta }\left(M\left(x\right)=y|s_{j},c,\theta \right)}P_{\theta }\left(c|s_{j},\theta \right)}\\ \; & =\sum_{c\in C}\frac{P_{M,\theta }\left(M\left(x\right)=y|s_{i},c,\theta \right)}{P_{M,\theta }\left(M\left(x\right)=y|s_{j},c,\theta \right)}P_{\theta }\left(c|s_{i},\theta \right)\\ \; & \leq \sum_{c\in C}e^{\epsilon }P_{\theta }\left(c|s_{i},\theta \right)=e^{\epsilon }. \end{array}$$ Reversing $s_{i}$ and $s_{j}$ shows the other inequality.    □

From Theorem 1, if we protect differential fairness per department, we obtain differential fairness and its corresponding theoretical economic and privacy guarantees in the university’s overall admissions, bounded by the ϵ of the most unfair department, *even in the case of a Simpson’s reversal*. If confounder variables are latent, we can attempt to infer them probabilistically in order to apply DFC. Alternatively, ($\epsilon_{2}-\epsilon_{1}$)-DF bias amplification can still be used to study the impact of an algorithm on fairness.

## 8. Properties of Differential Fairness

We now discuss the theoretical properties of our definitions.

### 8.1. Differential Fairness and Intersectionality

Differential fairness explicitly encodes protection of intersectional groups (Criterion B). For DF, we prove that this automatically implies fairness for *each of the protected attributes individually* (Criterion C), and indeed, *any subset* of the protected attributes. For example, if a loan approval mechanism M(x) is ϵ-DF in A=
*gender* × *race* × *nationality*, it is also ϵ-DF in, e.g., A=
*gender* by itself, or A=
*gender* × *nationality*. In other words, by ensuring fairness at the intersection of *gender*, *race*, and *nationality*, we also ensure the same degree of fairness between *genders* overall, between *gender*/*nationality* pairs overall, and so on. Here, ϵ is a worst case, and DF may also hold for lower values of ϵ.

**Lemma** **1.**
*The ϵ-DF criterion can be rewritten as: for any θ∈Θ, y∈Range(M),*

(10)
$$\begin{array}{cc} \; & \log\max_{s\in A:P(s|\theta )>0}P_{M,\theta }\left(M\left(x\right)=y|s,\theta \right)\\ \; & -\log\min_{s\in A:P(s|\theta )>0}P_{M,\theta }\left(M\left(x\right)=y|s,\theta \right)\leq \epsilon . \end{array}$$



**Proof.** The definition of ϵ-differential fairness is, for any θ∈Θ, y∈Range(M), $(s_{i},s_{j})\in A\times A$ where $P(s_{i}|\theta )>0$, $P(s_{j}|\theta )>0$,
(11)$$\begin{array}{cc} e^{-\epsilon }\leq \frac{P_{M,\theta }\left(M\left(x\right)=y|s_{i},\theta \right)}{P_{M,\theta }\left(M\left(x\right)=y|s_{j},\theta \right)} & \leq e^{\epsilon }. \end{array}$$Taking the log, we can rewrite this as:
(12)$$\begin{array}{cc} & -\epsilon \leq \log P_{M,\theta }\left(M\left(x\right)=y|s_{i},\theta \right)\\ \; & -\log P_{M,\theta }\left(M\left(x\right)=y|s_{j},\theta \right)\leq \epsilon . \end{array}$$The two inequalities can be simplified to:
(13)$$\begin{array}{cc} |\log P_{M,\theta }\left(M\left(x\right)=y|s_{i},\theta \right)-\log P_{M,\theta }\left(M\left(x\right)=y|s_{j},\theta \right)| & \leq \epsilon . \end{array}$$For any fixed θ and *y*, we can bound the left-hand side by plugging in the worst case over $\left(s_{i},s_{j}\right)$:
(14)$$\begin{array}{cc} \; & |\log P_{M,\theta }\left(M\left(x\right)=y|s_{i},\theta \right)-\log P_{M,\theta }\left(M\left(x\right)=y|s_{j},\theta \right)|\\ \; & \leq \log\max_{s:P(s|\theta )>0}P_{M,\theta }\left(M\left(x\right)=y|s,\theta \right)\\ \; & \;\;\;\;-\log\min_{s:P(s|\theta )>0}P_{M,\theta }\left(M\left(x\right)=y|s,\theta \right). \end{array}$$Plugging in this bound, which is achievable and hence is tight, the criterion is then equivalent to:
(15)$$\begin{array}{cc} \; & \log\max_{s:P(s|\theta )>0}P_{M,\theta }\left(M\left(x\right)=y|s,\theta \right)\\ \; & \;\;\;\;-\log\min_{s:P(s|\theta )>0}P_{M,\theta }\left(M\left(x\right)=y|s,\theta \right)\leq \epsilon . \end{array}$$   □

**Theorem** **2.*****(Intersectionality Property)*** *Let M be an ϵ-differentially fair mechanism in (A,Θ), $A=S_{1}\times S_{2}\times \ldots \times S_{p}$, and let $D=S_{a}\times \ldots \times S_{k}$ be the Cartesian product of a nonempty proper subset of the protected attributes included in A. Then M is ϵ-differentially fair in (D,Θ).*

**Proof.** Define $E=S_{1}\times \ldots \times S_{a-1}\times S_{a+1}\ldots \times S_{k-1}\times S_{k+1}\times \ldots \times S_{p}$, the Cartesian product of the protected attributes included in *A* but not in *D*. Then for any θ∈Θ, y∈Range(M),
$$\begin{array}{cc} \; & \log\max_{s\in D:P(s|\theta )>0}P_{M,\theta }\left(M\left(x\right)=y|D=s,\theta \right)\\ = & \log\max_{s\in D:P(s|\theta )>0}\sum_{e\in E}P_{M,\theta }\left(M\left(x\right)=y|E=e,s,\theta \right)P_{\theta }\left(E=e|s,\theta \right)\\ \leq & \log\max_{s\in D:P(s|\theta )>0}\sum_{e\in E}\max_{e^{\prime }\in E:P_{\theta }\left(E=e^{\prime }|s,\theta \right)>0}\\ \; & \;\;\;\;\;\;\;\;\;\;\;\left(P_{M,\theta }\left(M\left(x\right)=y|E=e^{\prime },s,\theta \right)\right)\times P_{\theta }\left(E=e|s,\theta \right)\\ = & \log\max_{s\in D:P(s|\theta )>0}\max_{e^{\prime }\in E:P_{\theta }\left(E=e^{\prime }|s,\theta \right)>0}P_{M,\theta }\left(M\left(x\right)=y|E=e^{\prime },s,\theta \right)\\ = & \log\max_{s^{\prime }\in A:P\left(s^{\prime }|\theta \right)>0}P_{M,\theta }\left(M\left(x\right)=y|s^{\prime },\theta \right) \end{array}$$By a similar argument,
(16)$$\begin{array}{c} \log\min_{s\in D:P(s|\theta )>0}P_{M,\theta }\left(M\left(x\right)=y|D=s,\theta \right)\geq \log\min_{s^{\prime }\in A:P\left(s^{\prime }|\theta \right)>0}P_{M,\theta }\left(M\left(x\right)=y|s^{\prime },\theta \right). \end{array}$$Applying Lemma 1, we hence bound ϵ in (D,Θ) as
(17)$$\begin{array}{cc} \; & \log\max_{s\in D:P(s|\theta )>0}P_{M,\theta }\left(M\left(x\right)=y|D=s,\theta \right)\\ \; & -\log\min_{s\in D:P(s|\theta )>0}P_{M,\theta }\left(M\left(x\right)=y|D=s,\theta \right)\\ \leq & \log\max_{s^{\prime }\in A:P\left(s^{\prime }|\theta \right)>0}P_{M,\theta }\left(M\left(x\right)=y|s^{\prime },\theta \right)\\ \; & -\log\min_{s^{\prime }\in A:P\left(s^{\prime }|\theta \right)>0}P_{M,\theta }\left(M\left(x\right)=y|s^{\prime },\theta \right)\leq \;\epsilon . \end{array}$$   □

This property is philosophically concordant with intersectionality, which emphasizes empathy with all overlapping marginalized groups. However, its benefits are mainly practical; in principle, one could protect all higher-level groups in γ-SF by specifying ∑j=1ppjKj binary indicator protected groups, where *K* is the number of values per protected attribute. This quickly becomes computationally and statistically infeasible. For example, Figure 7 counts the number of protected groups that must be explicitly considered under the two intersectional fairness definitions, in order to respect the intersectional fairness Criteria b and c. The intersectionality property (Theorem 2) implies that when the the bottom-level intersectional groups are protected (blue curve), differential fairness will automatically protect all higher-level groups (red curve). Since statistical parity subgroup fairness does not have this property, all of the groups and subgroups (red curve) must be protected explicitly with their own group indicators g(s). Although the number of bottom-level groups grows exponentially in the number of protected attributes, the total number of groups grows much faster, at the combinatorial rate of ∑j=1ppjKj.

### 8.2. Privacy Interpretation

The differential fairness definition and the resulting level of fairness obtained at any particular measured fairness parameter ϵ can be interpreted by viewing the definition through the lens of privacy. Differential fairness ensures that given the outcome, *an untrusted vendor/adversary can learn very little about the protected attributes of the individual*, relative to their prior beliefs, assuming their prior beliefs are in Θ:(18)$$e^{-\epsilon }\frac{P(s_{i}|\theta )}{P(s_{j}|\theta )}\leq \frac{P(s_{i}|M\left(x\right)=y,\theta )}{P(s_{j}|M\left(x\right)=y,\theta )}\leq e^{\epsilon }\frac{P(s_{i}|\theta )}{P(s_{j}|\theta )}.$$For example, if a loan is given to an individual, an adversary’s Bayesian posterior beliefs about their *race* and *gender* will not be substantially changed. Thus, the adversary will be unable to infer that “this individual was given a loan, so they are probably white and male.” Our definition thereby provides fairness guarantees when the user of M(x) is untrusted, cf. [7], by preventing subsequent discrimination, e.g., in retaliation to a fairness correction. Although DF is a *population-level* definition, it provides a privacy guarantee for *individuals*. The privacy guarantee only holds if θ∈Θ, which may not always be the case. Regardless, the value of ϵ may typically be interpreted as a privacy guarantee against a “reasonable adversary.” The privacy guarantee is inherited from *pufferfish*, a general privacy framework which DF instantiates [27].

### 8.3. Economic Guarantees

We also show that an economic guarantee for DF holds. An ϵ-differentially fair mechanism admits a disparity in expected utility of as much as a factor of exp(ϵ)≈1+ϵ (for small values of ϵ) between pairs of protected groups with $s_{i}\in A$, $s_{j}\in A$, for any utility function that could be chosen. For example, consider a loan approval process, where the utility of being given a loan is 1, and being denied is 0. Suppose the approval process is ln(3)-differentially fair. The process could then be three times as likely to award a loan to white men as to white women, and thus award white men three times the expected utility as white women. The proof follows the case of differential privacy [20]. Let $u\left(y\right):Range\left(M\left(x\right)\right)\to R_{\geq 0}$ be a utility function. Then:(19)$$\begin{array}{cc} E_{P_{M,\theta }}\left[u\left(y\right)|s_{i}\right] & =\int P_{M,\theta }\left(y|s_{i}\right)u\left(y\right)dy\\ \; & \leq \int e^{\epsilon }P_{M,\theta }\left(y|s_{j}\right)u\left(y\right)dy=e^{\epsilon }E_{P_{M,\theta }}\left[u\left(y\right)|s_{j}\right]. \end{array}$$Similarly, for $\left(\epsilon_{2}-\epsilon_{1}\right)$-DF bias amplification, M(x) admits at most an $\exp\left(\epsilon_{2}-\epsilon_{1}\right)\approx 1+\epsilon_{2}-\epsilon_{1}$ (for small values of $\epsilon_{2}-\epsilon_{1}$) multiplicative increase in the disparity of expected utility between pairs of protected intersections of groups with $s_{i}\in A$, $s_{j}\in A$, relative to the data-generating process M, for any utility function.

Note that the privacy and economic guarantees afforded by differential fairness (Equations (Equation 18) and (Equation 19), respectively) show that the definition is operationally significant, in the sense that the definition provides a guarantee on the harms due to the algorithm (or at least, its direct harms, neglecting its broader impacts within the complicated sociotechnical system in which it is embedded [22]). Impacted individuals and stakeholders can interpret the mechanism’s value of ϵ in terms of how its privacy and utility properties might impact them. This interpretability is a key advantage over competing definitions.

### 8.4. Generalization Guarantees

In order to ensure that an algorithm is truly fair, it is important that the fairness properties obtained on a dataset will extend to the underlying population. Kearns et al. [17] proved that empirical estimates of the quantities per group which determine subgroup fairness, $P_{M,\theta }\left(y=1|g\left(s\right)=1\right)P_{\theta }\left(g\left(s\right)=1\right)$, will be similar to their true values, with enough data relative to the VC dimension of the classification model’s concept class H. We state their result below.

**Theorem** **3.*****[17]’s Theorem 2.11 (SP Uniform Convergence).*** *Fix a class of functions H and a class of group indicators G. For any distribution P, let $S∼P^{m}$ be a dataset consisting of m examples $\left(x_{i},y_{i}\right)$ sampled i.i.d. from P. Then for any 0<δ<1, with probability 1−δ, for every h∈H and g∈G, we have:*(20)$$\begin{array}{cc} \; & \left|P(y=1|g(s)=1,h)P(g(s)=1\right)\\ \; & -P_{S}\left(y=1|g\left(s\right)=1,h\right)P_{S}\left(g\left(s\right)=1\right)|\\ \; & \leq \tilde{O}\left(\sqrt{\frac{\left(VCDIM(H)+VCDIM(G))\log m+\log(1/\delta \right)}{m}}\right). \end{array}$$

Here, $\tilde{O}$ hides logarithmic factors, and $P_{S}$ is the empirical distribution from the *S* samples. It is natural to ask whether a similar result holds for differential fairness. As Kearns et al. [17] note, the SF definition was chosen for statistical reasons, revealed in the above equation; the $P_{\theta }\left(g\left(s\right)=1\right)$ term in SF arises naturally in their generalization bound. For DF, we specifically avoid this term due to its impact on minority groups and must instead bound $P_{M,\theta }\left(y|s\right)$ per group s. For this case, we prove the following generalization guarantee.

**Theorem** **4.**
*Fix a class of functions H, which w.l.o.g. aim to discriminate the outcome y=1 from any other value, denoted as y=0. For any conditional distribution P(y,x|s) given a group s, let $S∼P^{m}$ be a dataset consisting of m examples $\left(x_{i},y_{i}\right)$ sampled i.i.d. from P(y,x|s). Then for any 0<δ<1, with probability 1−δ, for every h∈H, we have:*

(21)
$$\begin{array}{cc} \; & |P\left(y=1|s,h\right)-P_{S}\left(y=1|s,h\right)|\\ \; & \;\;\;\;\;\;\;\;\;\leq \tilde{O}\left(\sqrt{\frac{VCDIM(H)\log m+\log(1/\delta )}{m}}\right). \end{array}$$



**Proof.** Let $g(s^{\prime })=1$ when $s^{\prime }=s$ and 0 otherwise, and let $G=\{g\left(s^{\prime }\right)\}$. We see that G has a VC-dimension of 0. The result follows directly by applying Theorem 3 (Kearns et al. [17]’s Theorem 2.11) to H and G and considering the bound for the distributions *P* over (x,y) where $P(g\left(s^{\prime }\right)=1)=1$.    □

While SF has generalization bounds which depend on the overall number of data points, DF’s generalization guarantee requires that we obtain a reasonable number of data points for each intersectional group in order to accurately estimate ϵ-DF. This difference, the price of removing the minority-biasing term, should be interpreted in the context of the differing goals of our work and Kearns et al. [17], who aimed to **prevent fairness gerrymandering** by protecting every conceivable subgroup that could be targeted by an adversary.

In contrast, our goal is to **uphold intersectionality**, which simply aims to enact a more nuanced understanding of unfairness than with a single protected dimension such as *gender* or *race*. In practice, the consideration of two or three intersecting protected dimensions already improves the nuance of assessment. Sufficient data per intersectional group can often be readily obtained in such cases, e.g., Buolamwini and Gebru [5] studied the intersection of *gender* and *skin color* on fairness. Similarly, Kearns et al. [17] focus on the challenge of *auditing* subgroup fairness when the subgroups cannot easily be enumerated, which is important in the fairness gerrymandering setting. In contrast, in our intended applications of preserving intersectional fairness, the number of intersectional groups is often only between around $2^{2}$ (e.g., binary gender and race, as originally considered by [16]) to around $2^{5}$ (e.g., the Civil Rights Act of 1964 prohibits discrimination along lines of five attributes: race, color, religion, sex, and national origin).

## 9. Learning Algorithm

In this section we introduce simple and practical learning algorithms for differentially fair classifiers (*DF-Classifiers*). Our algorithms use the fairness cost as a regularizer to balance the trade-off between fairness and accuracy. We minimize, with respect to the classifier $M_{W}\left(x\right)$’s parameters **W**, a loss function *L* such as cross-entropy loss plus a penalty on unfairness which is weighted by a tuning parameter λ>0. In practice, we found that a warm start optimizing loss function *L* only for several “burn-in” iterations improves convergence.

The fairness penalty term $R_{X}\left(\epsilon \right)$ for training data X is designed as
(22)$$R_{W}\left(\epsilon \right)=max\left(0,\epsilon_{M_{W}\left(x\right)}-\epsilon_{1}\right),$$
where $\epsilon_{M_{W}\left(x\right)}$ is the ϵ-DF measures for $M_{W}\left(x\right)$. If $\epsilon_{1}$ is 0, this penalizes ϵ-DF, and if $\epsilon_{1}$ is the data’s ϵ, this penalizes bias amplification. Optimizing for bias amplification will also improve ϵ-DF, up to the $\epsilon_{1}$ threshold. We develop two approaches of learning algorithm that ensure the reliable and data-efficient estimation of $\epsilon_{M_{W}\left(x\right)}$ in the training phase.

### 9.1. Batch Method

We process all the training examples D=(x,y) simultaneously for batch DF (BDF) model $M_{W^{B}}\left(x\right)$. The learning objective for BDF becomes
(23)$$\min_{W^{B}}\left[\frac{1}{n_{D}}\sum_{i=1}^{n_{D}}L\left(f\left(x^{\left(i\right)};W^{B}\right),y^{\left(i\right)}\right)+\lambda R_{W^{B}}^{D}\left(\epsilon \right)\right],$$
where $R_{W^{B}}^{D}\left(\epsilon \right)=max\left(0,\epsilon_{M_{W^{B}}\left(x\right)}-\epsilon_{1}\right)$ represents the fairness penalty term in the batch method, and $f(x;W^{B})$ is the predicted output (analogous to P(y|x) in Equation (Equation 8)) for $n_{D}$ training examples. $\epsilon_{M_{W^{B}}\left(x\right)}$ is measured for the entire training data D using soft counts (Equation (Equation 8)) to make the objective differentiable. We train BDF using batch or deterministic gradient descent (GD) with learning step size schedule $\rho_{t}^{W^{B}}$ on the objective (Equation (Equation 23)). Pseudo-code to train the BDF model is given in Algorithm 1. In practice, we use *Adam* optimization [60] via backpropagation (BP) [61] and automatic differentiation (autodif) [62] with a fixed step size $\rho^{W^{B}}$. The main limitation of the BDF algorithm is that it involves calculating the relevant probabilities for all intersecting protected groups in every iteration, which is exponential in the number of top-level groups. To address this, we present a stochastic method in the following section.
**Algorithm 1:** Training Algorithm for Batch Differential Fair (BDF) Model**Require:** Train data D=(x,y)**Require:** Tuning parameter λ>0**Require:** Learning step size schedule $\rho_{1}^{W^{B}},\rho_{2}^{W^{B}},\cdots$**Require:** Randomly initialize model parameters $W^{B}$**Output:** BDF model $M_{W^{B}}\left(x\right)$For each burn-in iteration $t_{b}$:
-Apply update using batch GD with $\rho_{t_{b}}^{W^{B}}$:            $W^{B}:=W^{B}-\rho_{t_{b}}^{W^{B}}\frac{1}{n_{D}}\nabla_{W^{B}}\sum_{i=1}^{n_{D}}L\left(f\left(x^{\left(i\right)};W^{B}\right),y^{\left(i\right)}\right)$
For each iteration *t*:
-Compute $\epsilon_{M_{W^{B}}\left(x\right)}\leq \frac{\sum_{x\in D:A=s_{i}}f\left(x;W^{B}\right)+\alpha }{N_{s_{i}}+\left|Y\right|\alpha }\frac{N_{s_{j}}+\left|Y\right|\alpha }{\sum_{x\in D:A=s_{j}}f\left(x;W^{B}\right)+\alpha }\leq \epsilon_{M_{W^{B}}\left(x\right)}$-Apply update using batch GD with $\rho_{t}^{W^{B}}$:            $W^{B}:=W^{B}-\rho_{t}^{W^{B}}\frac{1}{n_{D}}\nabla_{W^{B}}\left[\sum_{i=1}^{n_{D}}L\left(f\left(x^{\left(i\right)};W^{B}\right),y^{\left(i\right)}\right)+\lambda R_{W^{B}}^{D}\left(\epsilon \right)\right]$                  //in practice, *Adam* optimization via BP and autodif with a fixed $\rho^{W^{B}}$

### 9.2. Stochastic Method

The stochastic DF (SDF) method $M_{W^{S}}\left(x\right)$ uses a minibatch or a small subset of all the training examples $D_{m}\in D$ for each iteration of learning. Pseudo-code to train SDF is given in Algorithm 2. The objective for SDF with minibatch size $n_{D_{m}}$ is equivalent to that of the the batch method:(24)$$\min_{W^{S}}\left[\frac{1}{n_{D_{m}}}\sum_{i=1}^{n_{D_{m}}}L\left(f\left(x^{\left(i\right)};W^{S}\right),y^{\left(i\right)}\right)+\lambda R_{W^{S}}^{D_{m}}\left(\epsilon \right)\right],$$
where $R_{W^{S}}^{D_{m}}\left(\epsilon \right)=max\left(0,\epsilon_{M_{W^{S}}\left(x\right)}-\epsilon_{1}\right)$ represents the fairness penalty term for a minibatch. However, the reliable estimation of $\epsilon_{M_{W^{S}}\left(x\right)}$ for a minibatch becomes statistically challenging due to data sparsity of intersectional groups [56]. For example, one or more missing intersectional groups for a minibatch is a typical scenario in the stochastic models that can lead to inaccurate estimation of intersectional fairness, i.e., ϵ-DF or γ-SF, which affects the training of the models.

We develop a stochastic approximation-based approach to address data sparsity in ϵ-DF estimation for training the SDF model. Our approach is mainly inspired by the online EM algorithm of [63]. It is a general-purpose method for learning latent variable models in an online setting that alternates between a standard M-step that maximizes the EM lower bound with respect to model parameters β and a stochastic expectation step that updates exponential family sufficient statistics Φ with an online average.
(25)$$\Phi :=\left(1-\rho_{t}\right)\Phi +\rho_{t}\hat{\Phi }\left(x^{\left(n+1\right)};\beta \right),$$
where $x^{\left(n+1\right)}$ is a new data point, $\rho_{t}$ is the step size at time step *t*, and $\hat{\Phi }\left(x^{\left(n+1\right)};\beta \right)$ is the estimation of the sufficient statistics based on the latest model parameters β. Inspired by the online EM algorithm, stochastic variational inference was proposed [64,65] to scale models such as topic models up to an enormous number of documents [66,67,68].
**Algorithm 2:** Training Algorithm for Stochastic Differential Fair (SDF) Model**Require:** Train data D=(x,y)**Require:** Tuning parameter λ>0**Require:** Constant learning step size schedule $\rho_{1}^{N^{G}},\rho_{2}^{N^{G}},\cdots$ and $\rho_{1}^{W^{S}},\rho_{2}^{W^{S}},\cdots$**Require:** Randomly initialize model parameters $W^{S}$ and count parameters $N_{y,s}^{G}$ and $N_{s}^{G}$**Output:** SDF model $M_{W^{S}}\left(x\right)$For each burn-in epoch:
-For each iteration $t_{b}$:
*Draw a minibatch $D_{m}\in D$*Apply update using SGD with $\rho_{t_{b}}^{W^{S}}$:      $W^{S}:=W^{S}-\rho_{t_{b}}^{W^{S}}\frac{1}{n_{D_{m}}}\nabla_{W^{S}}\sum_{i=1}^{n_{D_{m}}}L\left(f\left(x^{\left(i\right)};W^{S}\right),y^{\left(i\right)}\right)$For each epoch:
-For each iteration $t_{o}$ in the outer loop:
*For each iteration $t_{i}$ in the inner loop:
·Draw a minibatch $D_{m}\in D$·Empirically estimate ${\hat{N}}_{y,s}^{L}=\sum_{x\in D_{m}:A=s}f\left(x;W^{S}\right)$ and ${\hat{N}}_{s}^{L}$·Apply update: $N_{y,s}^{G}:=\left(1-\rho_{t_{i}}^{N^{G}}\right)N_{y,s}^{G}+\rho_{t_{i}}^{N^{G}}\frac{n_{D}}{n_{D_{m}}}{\hat{N}}_{y,s}^{L}$·Apply update: $N_{s}^{G}:=\left(1-\rho_{t_{i}}^{N^{G}}\right)N_{s}^{G}+\rho_{t_{i}}^{N^{G}}\frac{n_{D}}{n_{D_{m}}}{\hat{N}}_{s}^{L}$*Compute $\epsilon_{M_{W^{S}}\left(x\right)}\leq \frac{N_{y,s_{i}}^{G}+\alpha }{N_{s_{i}}^{G}+\left|Y\right|\alpha }\frac{N_{s_{j}}^{G}+\left|Y\right|\alpha }{N_{y,s_{j}}^{G}+\alpha }\leq \epsilon_{M_{W^{S}}\left(x\right)}$*Draw a minibatch $D_{m}\in D$*Apply update using SGD with $\rho_{t_{o}}^{W^{S}}$:         $W^{S}:=W^{S}-\rho_{t_{o}}^{W^{S}}\frac{1}{n_{D_{m}}}\nabla_{W^{S}}\left[\sum_{i=1}^{n_{D_{m}}}L\left(f\left(x^{\left(i\right)};W^{S}\right),y^{\left(i\right)}\right)+\lambda R_{W^{S}}^{D_{m}}\left(\epsilon \right)\right]$

To compute $\epsilon_{M_{W^{S}}\left(x\right)}$, we iterate over an inner loop $t_{i}$ for each outer loop $t_{o}$, with multiple minibatches drawn from training examples $D_{m}\in D$. First, we empirically estimate noisy expected counts per group ${\hat{N}}_{y,s}^{L}$ and ${\hat{N}}_{s}^{L}$ for each minibatch in the inner loop and update global expected counts per group $N_{y,s}^{G}$ and $N_{s}^{G}$ (shared within inner and outer loop iterations) with learning step size schedule $\rho_{t_{i}}^{N^{G}}$, which is typically annealed towards zero, as follows: (26)$$\begin{array}{c} N_{y,s}^{G}:=\left(1-\rho_{t_{i}}^{N^{G}}\right)N_{y,s}^{G}+\rho_{t_{i}}^{N^{G}}\frac{n_{D}}{n_{D_{m}}}{\hat{N}}_{y,s}^{L}, \end{array}$$(27)$$\begin{array}{c} N_{s}^{G}:=\left(1-\rho_{t_{i}}^{N^{G}}\right)N_{s}^{G}+\rho_{t_{i}}^{N^{G}}\frac{n_{D}}{n_{D_{m}}}{\hat{N}}_{s}^{L}, \end{array}$$
where ${\hat{N}}_{y,s}^{L}=\sum_{x\in D_{m}:A=s}f\left(x;W^{S}\right)$ are empirically estimated for each minibatch $D_{m}$ with respect to the latest $W^{S}$ (from the last outer loop iteration), while ${\hat{N}}_{s}^{L}$ is the total population per group for the corresponding minibatch. Note that it is too time-consuming to run the above inner loop for several minibatch examples. In practice, we found that one inner loop pass with a fixed learning step size $\rho^{N^{G}}$ is enough to train SDF model successfully, since $N_{y,s}^{G}$ and $N_{s}^{G}$ are continuously updated and shared in the outer loop iterations. Then, $\epsilon_{M_{W^{S}}\left(x\right)}$ is estimated in the outer loop via the posterior predictive distribution of a Dirichlet-multinomial (similar to Equation (Equation 7)) as
(28)$$\epsilon_{M_{W^{S}}\left(x\right)}\leq \frac{N_{y,s_{i}}^{G}+\alpha }{N_{s_{i}}^{G}+\left|Y\right|\alpha }\frac{N_{s_{j}}^{G}+\left|Y\right|\alpha }{N_{y,s_{j}}^{G}+\alpha }\leq \epsilon_{M_{W^{S}}\left(x\right)}.$$

Finally, the SDF model is trained using stochastic gradient descent (SGD) with step size schedule $\rho_{t_{o}}^{W^{S}}$ in the outer loop. In practice, we use *Adam* optimization algorithm on the objective in Equation (Equation 24) via BP and autodif with a fixed step size $\rho^{W^{S}}$. We also provide the pseudo-code to our more practical approach for the SDF model in Algorithm 3.
**Algorithm 3:** Practical Approach for Stochastic Differential Fair (SDF) Model**Require:** Train data D=(x,y)**Require:** Tuning parameter λ>0**Require:** Constant learning step size $\rho^{N^{G}}$ and $\rho^{W^{S}}$**Require:** Randomly initialize model parameters $W^{S}$ and count parameters $N_{y,s}^{G}$ and $N_{s}^{G}$**Output:** SDF model $M_{W^{S}}\left(x\right)$For each burn-in epoch:
-For each iteration $t_{b}$:
*Draw a minibatch $D_{m}\in D$*Apply *Adam* optimizer with $\rho^{W^{S}}$ via BP and autodif on the objective:            $\min_{W^{S}}\left[\frac{1}{n_{D_{m}}}\sum_{i=1}^{n_{D_{m}}}L\left(f\left(x^{\left(i\right)};W^{S}\right),y^{\left(i\right)}\right)\right]$For each epoch:
-For each iteration *t*:
*Draw a minibatch $D_{m}\in D$*Empirically estimate ${\hat{N}}_{y,s}^{L}=\sum_{x\in D_{m}:A=s}f\left(x;W^{S}\right)$ and ${\hat{N}}_{s}^{L}$*Apply update: $N_{y,s}^{G}:=\left(1-\rho^{N^{G}}\right)N_{y,s}^{G}+\rho^{N^{G}}\frac{n_{D}}{n_{D_{m}}}{\hat{N}}_{y,s}^{L}$*Apply update: $N_{s}^{G}:=\left(1-\rho^{N^{G}}\right)N_{s}^{G}+\rho^{N^{G}}\frac{n_{D}}{n_{D_{m}}^{t}}{\hat{N}}_{s}^{L}$*Compute $\epsilon_{M_{W^{S}}\left(x\right)}\leq \frac{N_{y,s_{i}}^{G}+\alpha }{N_{s_{i}}^{G}+\left|Y\right|\alpha }\frac{N_{s_{j}}^{G}+\left|Y\right|\alpha }{N_{y,s_{j}}^{G}+\alpha }\leq \epsilon_{M_{W^{S}}\left(x\right)}$*Apply *Adam* optimizer with $\rho^{W^{S}}$ via BP and autodif on the objective:            $\min_{W^{S}}\left[\frac{1}{n_{D_{m}}}\sum_{i=1}^{n_{D_{m}}}L\left(f\left(x^{\left(i\right)};W^{S}\right),y^{\left(i\right)}\right)+\lambda R_{W^{S}}^{D_{m}}\left(\epsilon \right)\right]$

### 9.3. Convergence Analysis for Stochastic Method

Our convergence analysis is conducted using the Robbins–Monro stochastic approximation (SA) [69] interpretation of the stochastic ϵ-DF estimation in the SDF algorithm. We have the following theorem:

**Theorem** **5.**
*If $0<\rho_{t}\leq 1$∀t, $\sum_{t=1}^{\infty }\rho_{t}=\infty$, and $\lim_{t\to \infty }\rho_{t}=0$, then in the limit as the number of steps t approaches infinity stochastic ϵ-DF estimation converges to a stationary point of the objective function for the SDF model.*


**Proof.** Consider the inner loop for stochastic ϵ-DF estimation in the SDF algorithm, with an update schedule for *t* steps. Noisy expected counts ${\hat{N}}_{y,s}^{L}$ and ${\hat{N}}_{s}^{L}$ are empirically estimated with respect to the SDF parameters $W^{S}$ and drawn minibatch $D_{m}\in D$ at the iteration step *t* of the inner loop. Given $W^{S}$, we do not need to maintain ${\hat{N}}_{y,s}^{L}$ and ${\hat{N}}_{s}^{L}$ between inner loop iterations. Thus, stochastic ϵ-DF estimation is operated only on the global expected count parameters $N_{y,s}^{G}$ and $N_{s}^{G}$. For each count parameter $c\in \{N_{y,s}^{G},N_{s}^{G}\}$, let $f_{c}\left(D,W^{S},\phi \right):\Phi_{c}\to \Phi_{c}$ be a mapping from a current value to the updated value after an iteration *t*, where $\Phi_{c}$ is the space of possible assignments for *c*.Let $\bar{\phi }=\left(N_{y,s}^{G},N_{s}^{G}\right)$ be an assignment of the expected count parameters, with ${\bar{\phi }}_{c}$ referring to a parameter *c*, and ${\bar{\phi }}^{\left(t\right)}$ be the *c* parameters at step *t*. Furthermore, let ${\hat{\bar{\phi }}}_{c}\left(D_{m}^{\left(t+1\right)},W^{S}\right)$ be the noisy estimate of $f_{c}\left(D,W^{S},\phi \right)$ based on the minibatch $D_{m}^{\left(t+1\right)}$ examined at step t+1. Finally, let $\xi^{\left(t+1\right)}={\hat{\bar{\phi }}}_{c}\left(D_{m}^{\left(t+1\right)},W^{S}\right)-f_{c}\left(D,W^{S},\phi^{\left(t\right)}\right)$ be the stochastic error made at step t+1, and observe that $E[\xi^{\left(t+1\right)}]=0$. We can rewrite the update for each *c* parameter as
(29)$$\begin{array}{cc} \; & {\bar{\phi }}_{c}^{\left(t+1\right)}=\left(1-\rho_{t+1}^{c}\right){\bar{\phi }}_{c}^{\left(t\right)}+\rho_{t+1}^{c}{\hat{\bar{\phi }}}_{c}\left(D_{m}^{\left(t+1\right)},W^{S}\right)\\ \; & ={\bar{\phi }}_{c}^{\left(t\right)}+\rho_{t+1}^{c}\left(-{\bar{\phi }}_{c}^{\left(t\right)}+{\hat{\bar{\phi }}}_{c}\left(D_{m}^{\left(t+1\right)},W^{S}\right)\right)\\ \; & ={\bar{\phi }}_{c}^{\left(t\right)}+\rho_{t+1}^{c}\left(f_{c}\left(D,W^{S},{\bar{\phi }}^{\left(t\right)}\right)-{\bar{\phi }}_{c}^{\left(t\right)}+{\hat{\bar{\phi }}}_{c}\left(D_{m}^{\left(t+1\right)},W^{S}\right)-f_{c}\left(D,W^{S},{\bar{\phi }}^{\left(t\right)}\right)\right)\\ \; & ={\bar{\phi }}_{c}^{\left(t\right)}+\rho_{t+1}^{c}\left(f_{c}\left(D,W^{S},{\bar{\phi }}^{\left(t\right)}\right)-{\bar{\phi }}_{c}^{\left(t\right)}+\xi^{\left(t+1\right)}\right). \end{array}$$In this form, we can see that iterating each of the *c* parameter updates over the inner loop to ϵ-DF estimation corresponds to the Robbins–Monro SA algorithm [69] for finding the zeros of $f_{c}\left(D,W^{S},\phi^{\left(t\right)}\right)-\phi_{c}^{\left(t\right)}$, i.e., the fixed points for $\phi_{c}$ that lead to the stationary point for $\epsilon_{M_{W^{S}}\left(x\right)}$ estimation in the SDF algorithm. Theorem 2.3 of [70] states that under the existence of a Lyapunov function along with a boundedness condition, this implies that such an SA algorithm will converge with step size schedules such as mentioned above. Note that a Lyapunov function can be viewed as an objective function in the absence of stochastic noise and the SA algorithm would improve the function monotonically with small enough steps in the direction of the updates. □

## 10. Experiments and Results

In this section, we performed an extensive experimental analysis of the proposed fairness metric in terms of fair learning algorithms and intersectionality. Our implementation’s source code is provided in the GitHub (https://github.com/rashid-islam/Differential_Fairness accessed on 10 April 2023).

### 10.1. Datasets

We performed all experiments with the following four small and large-scale datasets. *We recognize that using risk assessment datasets such as COMPAS or HMDA for fairness research is contested due to the biases inherent in criminal justice and financial systems [71]. Our results should be taken as an illustration of how our definition is operationally different rather than as an endorsement of using automated decison making for these applications.*

**COMPAS:** The COMPAS dataset regarding a system that is used to predict criminal recidivism, and which has been criticized as potentially biased [4]. We used *race* and *gender* as protected attributes. Gender was coded as binary. Race originally had six values, but we merged “Asian” and “Native American” with “other,” as all three contained very few instances. We predict “actual recidivism,” which is binary, within a 2-year period for 7.22 K individuals.**Adult:** The Adult 1994 U.S. census income data from the UCI ML-repository [72] consists of 14 attributes regarding work, relationships, and demographics for individuals, who are labeled according to whether their income exceeds USD 50,000/year (Predicted income used for consequential decisions such as housing approval may result in *digital redlining* [1].) pre-split into a training set of 32.56 K and a test set of 16.28 K instances. We considered *race*, *gender*, and *nationality* as the protected attributes. Gender was coded as binary. As most instances have U.S. nationality, we treat nationality as binary also between U.S. and “other.” The race attribute originally had five values, but we merged the “Native American” with “other,” as both contained very few instances.**HHP:** This is a medium-sized dataset which is derived from the Heritage Health Prize (HHP) milestone 1 challenge (https://www.kaggle.com/c/hhp accessed on 10 April 2023). The dataset contains information for 171.07 K patients over a 3-year period. Our goal is to predict whether the *Charlson Index*, an estimation of patient mortality, is greater than zero. Following Song et al. [73], we consider *age* and *gender* as the protected attributes, where there are nine possible *age* values and two possible *gender* values.**HMDA:** This is the largest dataset in our study, which is derived from the loan application register form for the Home Mortgage Disclosure Act (HMDA) [74]. The HMDA is a federal act approved in 1975 which requires mortgage lenders to keep records of information regarding their lending practices to create greater transparency and borrower protections in the residential mortgage market (https://www.ffiec.gov/hmda/ accessed on 10 April 2023). The downstream task is to predict whether the loan application is approved. We used *ethnicity*, *race*, and *gender* as protected attributes, where ethnicity and gender were coded as binary. The race attribute originally had five values such as “American-Indian/Alaska-Native,” “Asian,” “Black/African-American,” “Native-Hawaiian or Other Pacific-Islander,” and “White.” we merged “American-Indian/Alaska-Native,” “Asian,” and “Native-Hawaiian or Other Pacific-Islander” to the same category as “other” since all three contained very few instances. Note that we filtered out individuals who did not declare information, i.e., incomplete or missing, related to the protected and/or other attributes. After pre-processing, the HMDA data consist of 30 attributes regarding loan type, property type, loan purpose, etc., for 2.33 M individuals.

### 10.2. Fair Learning Algorithms

The goals of our experiments were to demonstrate the practicality of our *batch differential fair (BDF)* and *stochastic differential fair (SDF)* classifiers in learning intersectionally fair classifiers and to compare their behavior to baseline methods, especially with regards to minorities. Instead of Kearns et al. [17]’s algorithm, we trained subgroup fair classifiers as batch (*BSF*) and stochastic (*SSF*) methods using the same approach in Algorithms 1 and 3, respectively, replacing ϵ with γ in Equation (Equation 23), i.e., $R_{W^{B}}^{D}\left(\gamma \right)=max\left(0,\gamma_{M_{W^{B}}\left(x\right)}-\gamma_{1}\right)$, and Equation (Equation 24), i.e., $R_{W^{S}}^{D_{m}}\left(\gamma \right)=max\left(0,\gamma_{M_{W^{S}}\left(x\right)}-\gamma_{1}\right)$, respectively. This simplifies and speeds up learning to handle deep neural networks used in our experiments. We also compare the models with a demographic parity-based *p%-Rule* [75] classifier baseline, which generalizes the 80% rule to measure disparate impact toward one protected group. Since demographic parity, by definition, assumes binary protected attributes, we select the most marginalized bottom-level group as the protected group (*black women non-USA* for Adult, *black women* for COMPAS, *women* with *age ≥ 85* for HHP, and *black hispanic women for HMDA*), compared to its complement. A logistic regression or SVM-based algorithm subject to p%-Rule constraint was proposed by Zafar et al. [75], but the performance of their algorithm was very poor compared to our deep learning-based approach. Therefore, we implemented *batch p%-Rule classifier (BPR)* and *stochastic p%-Rule classifier (SPR)* using our DNN-based Algorithms 1 and 3, respectively. Since higher is better for p%-Rule, we modified Equation (Equation 23) as $R_{W^{B}}^{D}\left(p\%\right)=max\left(0,p{\%}_{1}-p{\%}_{M_{W^{B}}\left(x\right)}\right)$, and Equation (Equation 24) as $R_{W^{S}}^{D_{m}}\left(p\%\right)=max\left(0,p{\%}_{1}-p{\%}_{M_{W^{S}}\left(x\right)}\right)$ for BPR and SPR, respectively. We also considered deep learning-based typical classifiers in batch (*BT*) and stochastic (*ST*) methods that do not incorporate any fairness interventions with their objective functions. The summary of learning algorithms is given in Table 2.

#### 10.2.1. Experimental Settings

All classifiers were trained on a common neural network architecture via adaptive gradient descent optimization (Adam) with learning rate = 0.001 using PyTorch [76]. The configuration of the network was 3 hidden layers, 16 neurons in each layer with drop out probability 0.5, “relu” and “sigmoid” activations for the hidden and output layers, respectively. For all batch fair methods, we trained for 500 iterations, disabling the fairness penalties for an additional 50 “burn-in” iterations. For all stochastic fair methods, we trained for 5 epochs with an additional “burn-in” epoch for the large-scale HMDA dataset, while 100 epochs + 5 “burn-in” epochs were used for the other datasets. Note that we did not use any “burn-in” period for the typical models (both BT and ST), since a warm start is redundant here due to the absence of fairness interventions.

We split the COMPAS dataset into 60% train, 20% development, and 20% test set. For Adult, we used the pre-specified test set and held out 30% from the training data as the development set. We held out 10% and 5% from the HHP and HMDA datasets, respectively, as the test set, using the remainder for training. We further held out 10% and 5% from the HHP and HMDA training sets, respectively, as the development set for each dataset. Since HMDA is a very large dataset, it was not feasible in terms of runtime and memory to train the models in batch setting for the entime HMDA training set. Therefore, we trained the batch models on HMDA by randomly taking a subset with 10% of the HMDA training set.

We learned fair classifiers in several settings; we set the target thresholds (1) to perfect fairness in terms of the corresponding fairness metric, $\epsilon_{1}$ = 0.0, $\gamma_{1}$ = 0.0, $p{\%}_{1}$ = 100.0% for DF-Classifiers (BDF and SDF), SF-Classifiers (BSF and SSF), and p%-Rule classifiers (BPR and SPR), respectively, and (2) to penalize bias amplification by the algorithm, by setting the thresholds to $\epsilon_{1}$ = $\epsilon_{data}$ and $\gamma_{1}$ = $\gamma_{data}$ for DF-Classifiers and SF-Classifiers, respectively. Finally, to protect the 80% rule, we set $\epsilon_{1}$ = −log0.8=0.2231 for DF-Classifiers. Since there is no straightforward way to enforce the 80% rule for SF-Classifier, it was not considered in this analysis. To evaluate the predictive performance of the classifiers, we compute *Accuracy*, *F1 score*, and *ROC AUC* for the held-out data. Finally, we compute *ϵ-DF*, *γ-SF*, *p%-Rule*, *bias amplification* on held-out data (using Equation (Equation 8)) to evaluate the classifiers in terms of fairness.

#### 10.2.2. Fairness and Accuracy Trade-Off

Fair learning algorithms divert the learning objective from accuracy only to both accuracy and fairness, which may hurt the predictive performance of the models. The tuning parameter λ allows the stakeholders to balance between fairness and accuracy. We chose λ for all fair learning algorithms via rigorous grid search on the development set based on a pre-defined rule: *select λ that provides the fairest (under the corresponding fairness metric, e.g., ϵ for DF-Classifiers, γ for SF-Classifiers) model on the development set, allowing up to 5% degradation in accuracy from the typical classifier*. Since fairness is a multi-stakeholder issue, the amount of slack tolerance on the accuracy can be amended based on the stakeholders’ preferences when deploying these methods in practice.

In Figure 8, we show the impact of the tuning parameter λ on the accuracy and corresponding fairness metric for the SDF and SSF classifiers on the development set of Adult and HHP datasets. *Black circles* in the plots represent the SDF or SSF model corresponding to different λ values (larger to smaller from left to right), while *red square* is the stochastic typical classifier (ST) (no fairness interventions in the objective). Larger λ incorporated with the corresponding fairness penalty for the SDF or SSF model allows us to achieve more fairness, but with greater loss in predictive performance, while smaller λ has the opposite impact on the models’ output. So, λ needs to be chosen as a trade-off between accuracy and fairness. Parallel *dotted lines* represent the valid region of 5% degraded accuracy from the ST model, and *green asterisk* is the fairest SDF or SSF model, in terms of their corresponding fairness metric, within the valid region for a particular λ value. According to our pre-defined rule, these λ values (*green arrow*) are then selected for SDF and SSF models. The trade-off between fairness and accuracy was similar for other datasets and is provided in the Appendix A (see Figure A1).

#### 10.2.3. Comparative Analysis for Batch and Stochastic Methods

In this experiment, we compare the accuracy and fairness for the batch and stochastic fair models (DF-Classifiers and SF-Classifiers) in terms of the training algorithm’s runtime. Stochastic fair models are scalable to very large data due their minibatch setting. For example, we trained the stochastic models on the entire HMDA training data (largest dataset in our study) without any additional complexity, while the batch models were trained on a small subset of the HMDA training data due to extremely high memory requirements and runtime. The purpose of this analysis is to show that the noisy updates used to train stochastic fair models do not introduce any additional harm to the accuracy or fairness of the models as we estimate the fairness penalty using noisy updates of the empirical counts (see Equations (Equation 26) and (27)).

Figure 9 shows ϵ-DF and accuracy in terms of runtime for the BDF and SDF models on the development set of Adult and HHP datasets. The *red X* indicates the end of burn-in iterations and burn-in epochs for the BDF and SDF models, respectively. Recall that, in the case of these two datasets, we disabled the fairness penalty for 50 burn-in iterations and 5 burn-in epochs in the training process of BDF and SDF models, respectively. In the SDF model, the burn-in epochs were enough to obtain a model that shows similar performance to the typical model (high accuracy and high ϵ-DF), while the model started declining in accuracy and improving ϵ-DF (lower is better) after burn-in due to the fairness interventions in the objective. Due to the fewer burn-in iterations in the training of the BDF model, we found that the model performed with relatively lower accuracy and less extreme ϵ-DF, since the predicted probability of the outcomes across protected groups were approximately similar. After burn-in, the BDF model improved the predictive performance and also maintained a low ϵ-DF from imposing the fairness penalty in the objective. Note that λ was pre-selected based on our pre-defined rule, as discussed in Section 10.2.2. Since both the BDF and SDF models were constrained to comply with the pre-defined rule for λ selection, after a certain period of runtime, both models converged to approximately similar solutions for both datasets (Adult in Figure 9a and HHP in Figure 9b) in terms of ϵ-DF and accuracy. We show a similar trend in Figure 10 for BSF and SSF models by comparing the γ-SF and accuracy with respect to the runtime. We conclude that stochastic models do not cause any additional harm to the model’s fairness or predictive performance comparing to the batch methods. See the appendix for similar results with the other datasets (Figure A2 and Figure A3).

#### 10.2.4. Performance for Fair Classifiers

In this experiment, we evaluate the accuracy-based performance and fairness metrics for batch and stochastic classifiers on the unseen test set of all datasets. Table 3 compares the batch methods on the Adult and COMPAS datasets. Both the BDF and BSF classifiers were able to **substantially improve their fairness metrics over the BPR and BT classifiers, with modest costs in accuracy, F1 score, and ROC AUC**, and the trade-off varied roughly monotonically in the target value $\epsilon_{1}$ or $\gamma_{1}$. The BDF model with $\epsilon_{1}=0$ improved from ϵ=0.743 to ϵ=0.205 on COMPAS data without any loss in accuracy, while also improving all other fairness metrics substantially. Surprisingly, BDF (and also the BPR classifier) slightly improved accuracy on this data compared to BT. This counter-intuitive result is presumably due to the regularization behavior of the fairness penalty on the objective, which can sometimes lead fair models to reduce overfitting to some degree compared to the typical model, a phenomenon which we observed in our previous work on the equitable allocation of healthcare resources [12]. On Adult, it improved from ϵ=1.591 to ϵ=0.305, corresponding to a worst-case difference in utility between groups of a factor of $e^{\epsilon }\approx 1.4$ (Equation (Equation 19)), with a loss of 3.6 percentage points in accuracy. In terms of γ and p%-Rule, the BSF model with $\gamma_{1}=0$ showed the highest improvement on the Adult data.

We found a similar trend in performance for stochastic methods. Table 4 compares the accuracy and fairness measures for the stochastic methods on the HHP and HMDA datasets. Similar to batch methods, SDF and SSF substantially improve fairness over the SPR and ST classifiers with modest costs in accuracy-based measures. On the HHP dataset, the SDF method with $\epsilon_{1}=0$ improved from ϵ=1.791 to ϵ=0.21, corresponding to a worst-case difference in utility between groups of a factor of $e^{\epsilon }\approx 1.2$, with an accuracy loss of 4.2 percentage points. SDF also showed the best improvement on this dataset in terms of γ (improved from γ=0.022 to γ=0.004). In the case of HMDA, it improved from ϵ=0.673 to ϵ=0.24 with a loss of 4.4 percentage points in accuracy. When BDF and SDF were trained to protect the 80% rule (i.e., $\epsilon_{1}=0.2231$) or to prevent bias amplification (i.e., $\epsilon_{1}=\epsilon_{data}$), the fairness metrics were improved with relatively little reduction in accuracy.

Our overall conclusions from Table 3 and Table 4 are as follows: the DF-Classifiers (BDF and SDF) typically had comparable accuracy under the same settings compared to the SF-Classifiers (BSF and SSF) and p%-Rule Classifiers (BPR and SPR), while the **DF-Classifiers often greatly improved γ-SF and p%-Rule, but the SF- and p%-Rule Classifiers had modest improvements, or even increases, in ϵ-DF**.

An important goal of this work was to address fairness for minority groups. In Figure 11 and Figure 12, we report the “per-group fairness”, defined as Equation (Equation 3) (γ-SF) and Equation (Equation 2) (ϵ-DF) with the group held fixed, versus the group’s probability (i.e., size) for batch and stochastic models, respectively. Both the DF-Classifiers and SF-Classifiers generally improved their corresponding unfairness per group over the Typical-Classifiers (BT or ST). On the other hand, similar to Figure 4, the γ-SF metric assigns the highest per-group γ-SF values, which determines overall fairness, to large groups, so **minority groups were not able to influence the overall γ-SF**. For example, overall γ-SF is determined by a top-level group (*red “X"*) for BSF on both Adult and COMPAS datasets and SSF on both HHP and HMDA datasets. This was not the case **for the ϵ-DF metric, where groups of various sizes had similarly high per-group ϵ-DF values**. For example, the overall ϵ-DF is determined by an intersectional subgroup (*blue “X"*) regardless of the subgroup’s size for both the BDF and SDF models on all the example datasets. Furthermore, the **DF-Classifiers improved the per-group fairness under both metrics for groups of all sizes, while the SF-classifiers made per-group γ-SF and ϵ-DF worse for very small groups**. Our overall conclusion is that the *DF-Classifiers are able to achieve intersectionally fair classification with minor loss in performance, while providing greater protection to minority groups than when enforcing SF*. Similar experimental results on the other datasets are provided in Appendix A (see Table A1 and Table A2 and Figure A4 and Figure A5).

### 10.3. Inequity of Fairness Measures

We have seen that the γ-SF metric downweights the consideration of minorities (cf. Figure 4, Figure 11 and Figure 12). In this experiment, we quantify the resulting inequity of fairness consideration using the *Gini coefficient* [77], a commonly used measure of statistical dispersion which is often used to represent the inequity of income. We calculate the *Gini coefficient G of a fairness metric F* as
(30)$$G=\frac{1}{2\mu }\sum_{i=1}^{n}\sum_{j=1}^{n}P\left(s_{i}\right)P\left(s_{j}\right)\left|F_{s_{i}}-F_{s_{j}}\right|,$$
where $\mu =\sum_{i=1}^{n}F_{s_{i}}P\left(s_{i}\right)$ and $P(s_{i})$ is the fraction of population belonging to the *i*th intersectional group, while $F_{s_{i}}$ represents the fairness measure (i.e., per-group ϵ or γ) of that group. For a fixed algorithm and data distribution, a fairness metric with a smaller Gini coefficient distributes its (un)fairness consideration more equitably across the population, which is typically desirable.

Table 5 shows a comparison of *G* values for the ϵ-DF and γ-SF metrics on all datasets. Both fairness metrics are measured for the labeled dataset (i.e., $\epsilon_{Data}$) as well as for a logistic regression (LR) classifier (i.e., $\epsilon_{LR}$) trained on the same dataset. In all the experiments, the *G* value for ϵ-DF is much lower compared to γ-SF’s *G* value. Thus, ϵ-DF was observed to provide a more equitable distribution of its per-group fairness measurements, presumably due to its more inclusive treatment of minority groups.

### 10.4. Evaluation of Intersectionality Property

In our final experiment (Table 6), we studied the ability of γ-SF to preserve the intersectionality property shown for ϵ-DF in Theorem 2, by measuring fairness with different sets of protected attributes on all datasets.

The property is violated if removing a protected attribute increases the metric. As expected, ϵ-DF obeyed the intersectionality property for all datasets, but γ-SF violated it as γ for *gender* > γ for *race* × *gender* (COMPAS); γ for *gender* > γ for *gender* × *nationality* (Adult); γ for *age* > *gender* × *age* (HHP); and γ for *race* × *ethnicity * > γ for *race* × *gender* × *ethnicity* (HMDA).

## 11. Discussion

In this work, we introduced fairness measures for AI/ML algorithms and data with regard to multiple protected attributes. Our proposed *differential fairness (DF)* metrics are informed by the framework of intersectionality, which analyzes how interlocking systems of power and oppression affect individuals along overlapping dimensions including race, gender, sexual orientation, class, and disability. DF has a particularly elegant intersectionality property that protects higher-level groups (e.g., *women*) automatically by protecting intersectional subgroups (*Black women*). There are several other attractive properties of our formalism and we provided proofs for the privacy, economic, and generalization properties. DF is *lightweight*, in that it does not require the specification or estimation of a causal model [54] or a latent model of risk distributions [43]. It is nevertheless able to make use of a probabilistic model of the data, when available, but does not require one. While no fairness definition is suitable in all contexts, due to these properties we recommend the use of differential fairness when an intersectional measure of fairness is appropriate.

It has been recently shown that many fairness notions can be captured for fair classifications as a combination of equality constraints between protected groups using Lagrangian dual approaches [78,79,80]. However, these methods were developed to enforce group fairness notions and cannot be directly adapted to learn fairness in an intersectional context, especially for learning in stochastic settings, due to the data sparsity of multi-dimensional protected attributes. We developed simple and practical learning algorithms which enforce DF, thereby ensuring fairness in a manner which behaves sensibly for any subset of the set of protected attributes. Our experimental results on multiple benchmark datasets demonstrate that the proposed algorithms substantially improve DF metrics over the baseline models, with modest costs in accuracy. Furthermore, unlike the baseline models, our algorithms often greatly improve other group-level (p%-Rule) and subgroup-level (SF) fairness metrics. Finally, we showed that our DF metric-based methods are able to achieve intersectionally fair classification with little loss in predictive performance, while providing greater protection to minority groups (in terms of population size per group) compared to SF metric-based methods.

However, the measurement of fairness becomes statistically challenging in the intersectional setting due to data sparsity, which increases rapidly in the number of dimensions (e.g., *gender*, *race*, *age*, etc.), and in the values per dimension (e.g., *race* attributes may include *black*, *white*, *asian*, etc.). In this paper, we addressed this by developing a learning algorithm for the reliable and stochastic approximation-based estimation of intersectional fairness penalty, incorporated with the objective function of deep neural networks. In our previous work [56], we proposed using hierarchical Bayesian probabilistic modeling to manage uncertainty of the intersectional fairness measures via MCMC or variational inference. It would be interesting to extend our approach to use this fully Bayesian uncertainty estimation strategy during training, although it would be costly to compute and differentiate MCMC samples or a variational posterior per minibatch for deep learning-based methods trained via stochastic gradient descent.

In other future work, differential privacy has been extended and generalized in various ways, including notably its generalization to Rényi differential privacy [21]. It would be interesting to investigate analogous generalizations for differential fairness. Another potential extension of the definition which is important for applications is a version that protects error rates as opposed to outcomes, following [8].

More generally, an important aspect of this work is to integrate research on fairness and bias issues in our society from disciplines such as the humanities, social science, and law, in our case regarding intersectionality, with technical methods for fair AI/machine learning. AI fairness is not a purely technical issue, and hence the field of computer science cannot solve it alone. There is an urgent need for more research on interdisciplinary approaches and perspectives on AI fairness, and this work is one small step in that direction. A special issue on this topic recently published in the IEEE Data Engineering Bulletin, edited by two of the authors of this paper, also aims to promote this emerging direction of research [81].

## 12. Conclusions

In this research we took an interdisciplinary approach to fairness in AI and machine learning systems by developing technical methods to implement intersectionality, a perspective on fairness and bias arising from the humanities, law, and the social sciences disciplines. We introduced three AI fairness definitions satisfying intersectional fairness desiderata, *differential fairness* and its *bias amplification* and *confounder-aware* counterparts, and proved their attractive properties regarding law, privacy, economics, and statistical learning. Our theoretical results show that differential fairness operationalizes fairness in a meaningful and interpretable way, as it quantifies the potential real-world economic and privacy harms due to the algorithm. We developed learning algorithms to enforce our criteria. We further addressed the data sparsity problem of intersectional fairness estimation which arises in SGD minibatches, using a novel intersectionally fair stochastic learning algorithm. In extensive experiments, we showed that our criteria can be practically attained, and that they behave more equitably with regard to minority groups than subgroup fairness. Going forward, we argue that it is essential to deeply integrate research from the humanities and social science on fairness and bias issues with work from the field of computer science, in order to holistically address both the human and algorithmic sides of the AI fairness equation. This work is a step in that direction, but much more remains to be done.

## Figures and Tables

**Figure 1 entropy-25-00660-f001:**
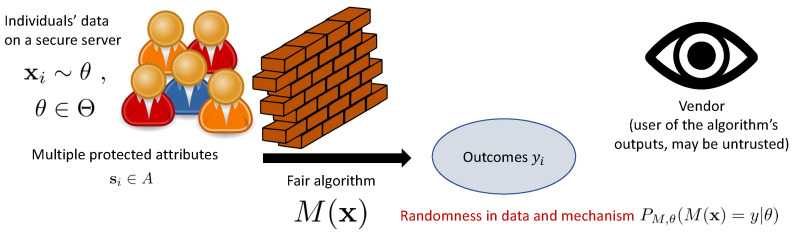
Diagram of the setting for the proposed differential fairness criterion.

**Figure 2 entropy-25-00660-f002:**
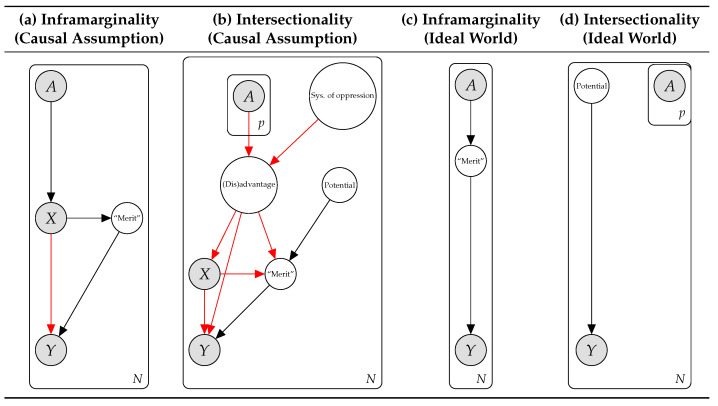
Implicit causal assumptions (**a**,**b**) and values-driven ideal-world scenarios (**c**,**d**) for infra-marginality and intersectionality notions of fairness. Here, *A* denotes protected attributes, *X* observed attributes, *Y* outcomes, *N* individuals, *p* number of protected attributes. Red arrows denote potentially unfair causal pathways, which are removed to obtain the ideal-world scenarios (**c**,**d**). The above summarizes broad strands of research; individual works may differ.

**Figure 3 entropy-25-00660-f003:**
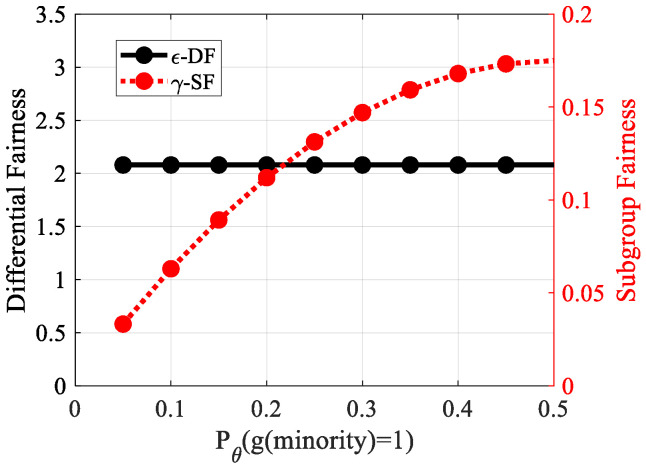
Toy example: probability of the “positive” class is 0.8 for a majority group, 0.1 for a minority group, varying $P_{\theta }\left(g\left(minority\right)=1\right)$.

**Figure 4 entropy-25-00660-f004:**
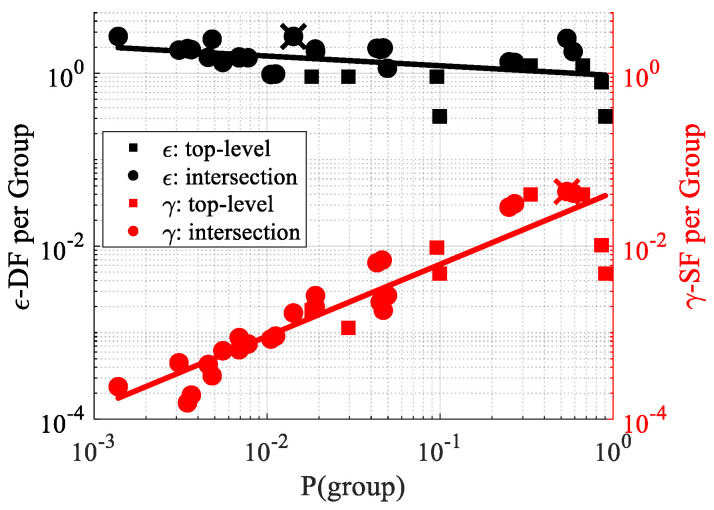
“Per-group” γ-SF and our proposed ϵ-DF vs. probability (i.e., size) of groups, UCI Adult dataset. Circles: intersectional subgroups (e.g., Black women of USA). Squares: top-level groups (e.g., men). Fitted least squares line demonstrates the overall trend between fairness measures and group size. Largest per-group ϵ-DF (*black “X"*) and γ-SF (*red “X"*) determines the overall ϵ-DF and γ-SF, respectively.

**Figure 5 entropy-25-00660-f005:**
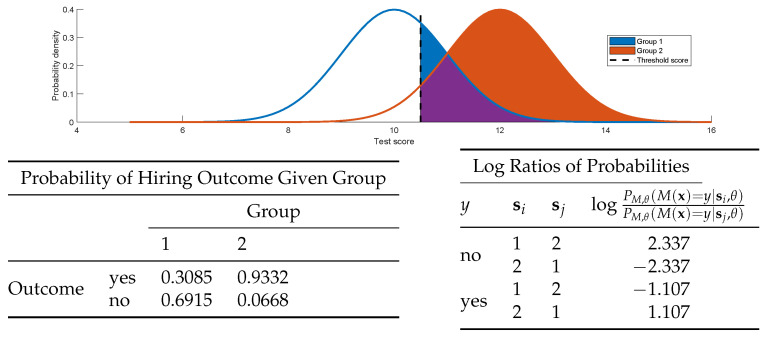
Worked example of differential fairness from Section 5. The calculations above show that ϵ=2.337.

**Figure 6 entropy-25-00660-f006:**
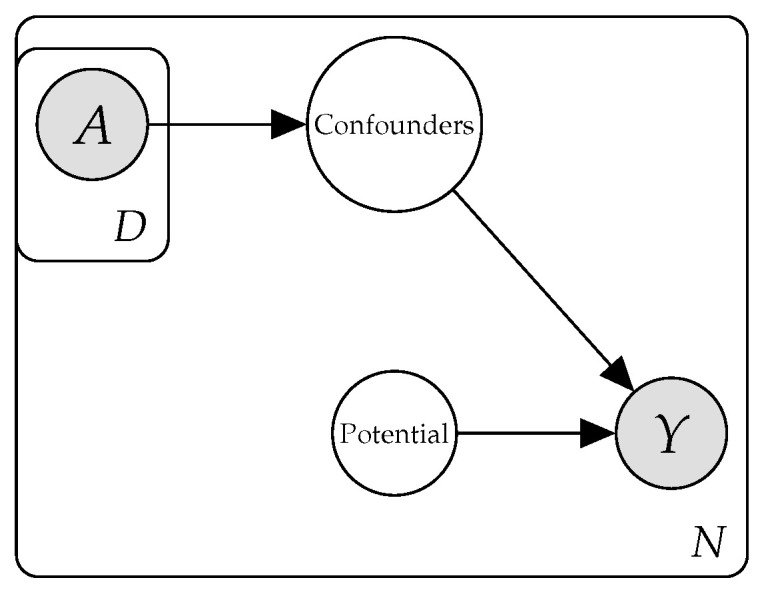
Ideal-world intersectional fairness with confounder variables. Disparity in overall outcomes between protected groups may be considered legitimate.

**Figure 7 entropy-25-00660-f007:**
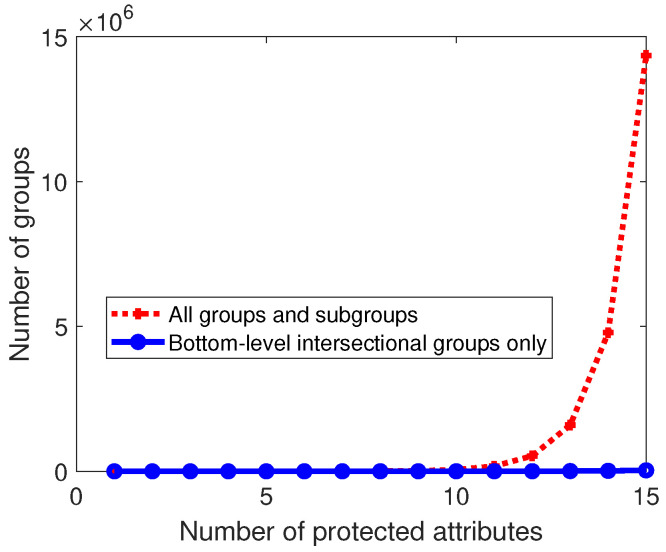
The number of groups and intersectional subgroups to protect when varying the number of protected attributes, with 2 values per protected attribute.

**Figure 8 entropy-25-00660-f008:**
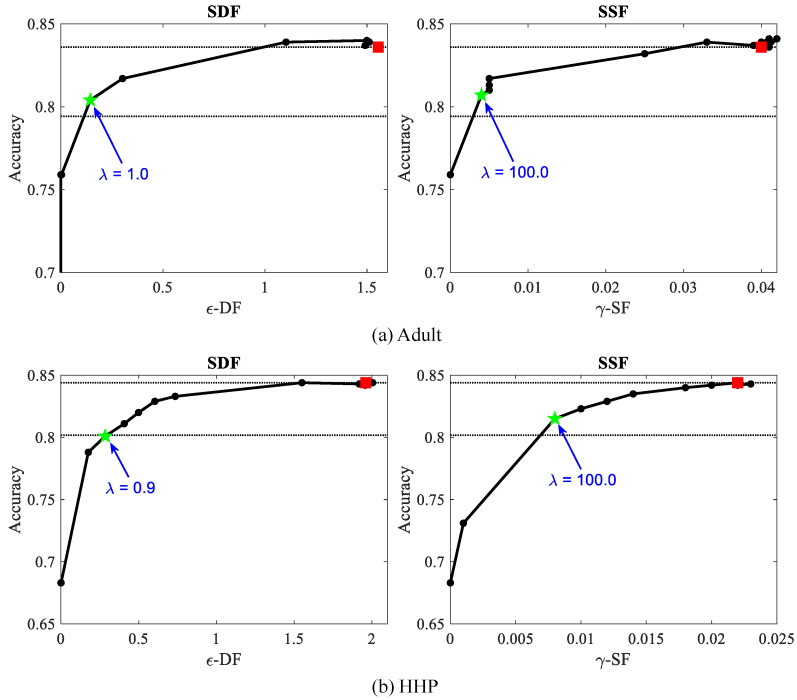
Fairness and accuracy trade-off plots for stochastic DF (SDF) and SF (SSF) models on the development set of (**a**) Adult and (**b**) HHP datasets. *Red square:* stochastic typical (ST) classifier that does not incorporate fairness penalty in the objective; *dotted lines:* indicate the valid region of 5% degraded accuracy from ST; *black circles*: fair models (SDF or SSF) correspond to different λ values (larger to smaller from left to right); *green asterisk:* fairest model within the valid region for a particular λ value.

**Figure 9 entropy-25-00660-f009:**
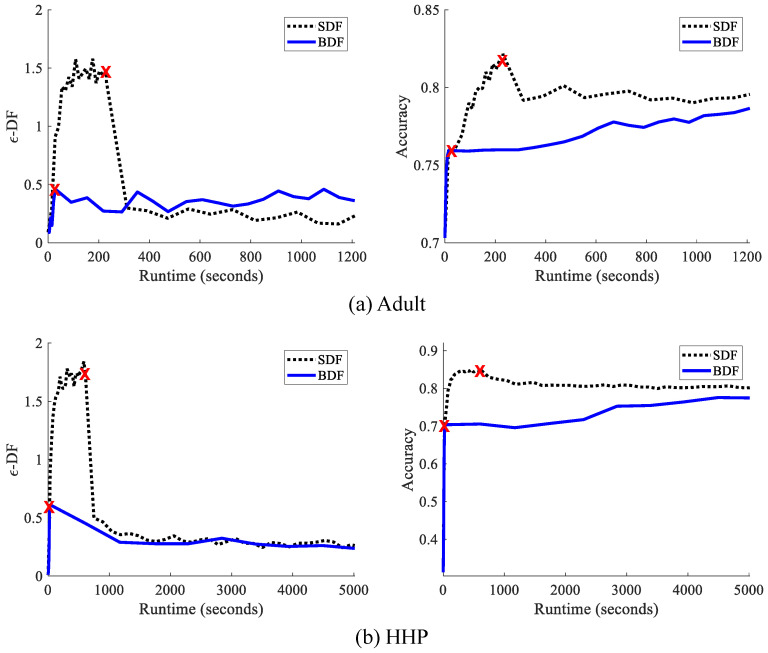
Comparison of ϵ-DF and accuracy in terms of runtime for the batch differential fair (BDF) and stochastic differential fair (SDF) models on the development set of (**a**) Adult and (**b**) HHP datasets. *Red X*: end of burn-in (disabling the fairness interventions) iterations and epochs for the BDF and SDF models, respectively.

**Figure 10 entropy-25-00660-f010:**
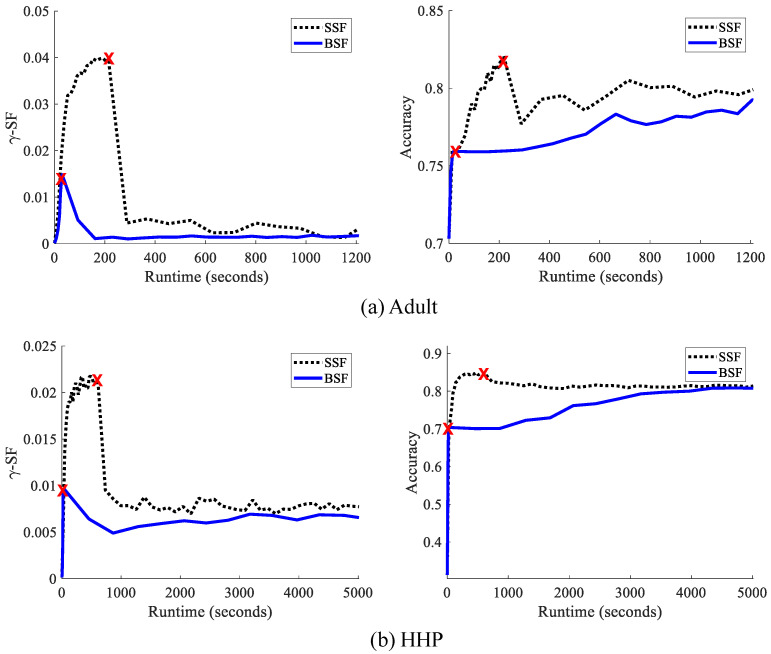
Comparison of γ-SF and accuracy in terms of runtime for the batch subgroup fair (BSF) and stochastic subgroup fair (SSF) models on the development set of (**a**) Adult and (**b**) HHP datasets. *Red X*: end of burn-in (disabling the fairness interventions) iterations and epochs for the BSF and SSF models, respectively.

**Figure 11 entropy-25-00660-f011:**
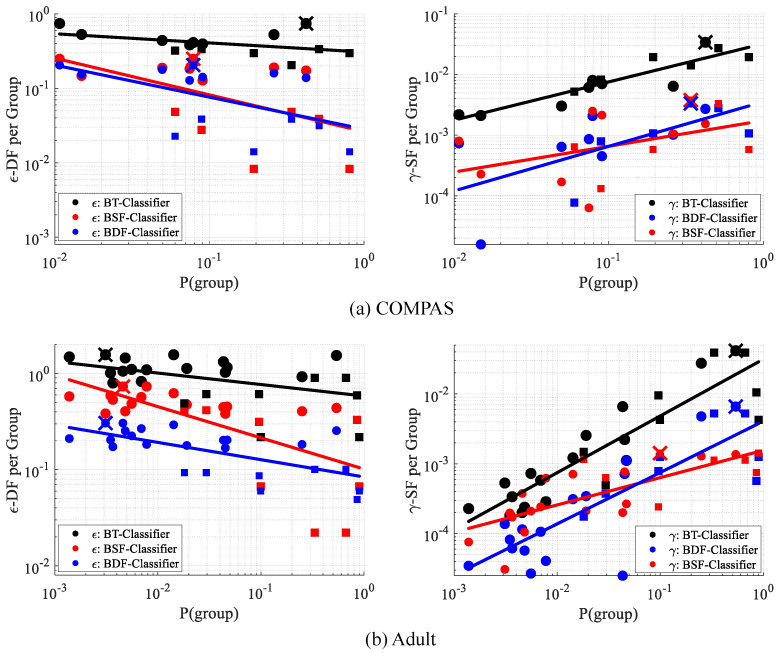
Per-group measurements of ϵ-DF and γ-SF for the batch differential fair (BDF), batch subgroup fair (BSF), batch p%-Rule (BPR), and batch typical (BT) classifiers vs. group size (probability), (**a**) COMPAS and (**b**) Adult datasets, calculated using Equation (Equation 3) (γ-SF) and Equation (Equation 2) (ϵ-DF) with the group held fixed. *Circles:* intersectional subgroups. *Squares:* top-level groups. Fitted least squares line demonstrates the overall trend between fairness measures and group size. Largest per-group ϵ-DF and γ-SF, indicated using *“X"*, determines the overall ϵ-DF and γ-SF, respectively.

**Figure 12 entropy-25-00660-f012:**
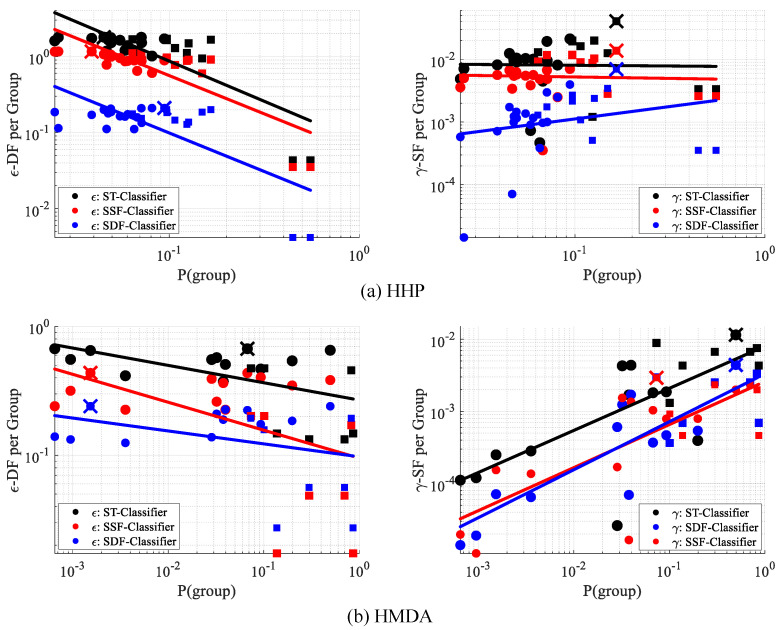
Per-group measurements of ϵ-DF and γ-SF for the stochastic differential fair (SDF), stochastic subgroup fair (SSF), stochastic p%-Rule (SPR), and stochastic typical (ST) classifiers vs. group size (probability), (**a**) HHP and (**b**) HMDA datasets, calculated using Equation (Equation 3) (γ-SF) and Equation (Equation 2) (ϵ-DF) with the group held fixed. *Circles:* intersectional subgroups. *Squares:* top-level groups. Fitted least squares line demonstrates the overall trend between fairness measures and group size. Largest per-group ϵ-DF and γ-SF, indicated using *“X"*, determines the overall ϵ-DF and γ-SF, respectively.

**Table 1 entropy-25-00660-t001:** Intersectional example: Simpson’s paradox.

Probability of Being Admitted to University X				
		**Gender**		
		* **A** *	* **B** *	**Overall**
Race	1	$\frac{81}{87}\;\left(0.931\right)$	$\frac{234}{270}\;\left(0.867\right)$	$\frac{315}{357}\;\left(0.882\right)$
	2	$\frac{192}{263}\;\left(0.730\right)$	$\frac{55}{80}\;\left(0.688\right)$	$\frac{247}{343}\;\left(0.720\right)$
	Overall	$\frac{273}{350}\;\left(0.780\right)$	$\frac{289}{350}\;\left(0.826\right)$	

**Table 2 entropy-25-00660-t002:** Summary of learning algorithms.

Abbreviation	Definition	Description
BT	Batch Typical (vanilla) model	Train deep learning classifier with batch method without any fairness interventions
BPR	Batch p%-Rule model	Train deep learning classifier with batch method that enforces p%-Rule criteria
BSF	Batch Subgroup Fair model	Train deep learning classifier with batch method that enforces γ-SF criteria
BDF	Batch Differential Fair model	Train deep learning classifier with batch method that enforces proposed ϵ-DF criteria
ST	Stochastic Typical (vanilla) model	Train deep learning classifier with stochastic method without any fairness interventions
SPR	Stochastic p%-Rule model	Train deep learning classifier with stochastic method that enforces p%-Rule criteria
SSF	Stochastic Subgroup Fair model	Train deep learning classifier with stochastic method that enforces γ-SF criteria
SDF	Stochastic Differential Fair model	Train deep learning classifier with stochastic method that enforces proposed ϵ-DF criteria

**Table 3 entropy-25-00660-t003:** Comparison of batch differential fair (BDF), batch subgroup fair (BSF), batch p%-Rule (BPR), and batch typical (BT) classifiers on the test set of COMPAS and Adult datasets ($\epsilon_{1}=0.2231$ is the 80% rule). *Higher is better* for measures with ↑; *lower is better* for measures with ↓.

COMPAS Dataset								
**Models**		**BDF-Classifier**			**BSF-Classifier**		**BPR-Classifier**	**BT-Classifier**
		$\epsilon_{1}=0.0$	$\epsilon_{1}=0.2231$	$\epsilon_{1}=\epsilon_{\mathrm{data}}$	$\gamma_{1}=0.0$	$\gamma_{1}=\gamma_{\mathrm{data}}$		
Tuning Parameter	λ	0.800	5.000	50.000	25.000	100.000	0.010	X
Performance Measures	Accuracy ↑	0.672	**0.674**	0.665	0.660	0.667	**0.674**	0.671
	F1 Score ↑	0.560	0.580	0.580	0.560	0.580	0.620	**0.630**
	ROC AUC ↑	0.700	0.698	0.687	**0.701**	**0.701**	0.700	0.686
Fairness Measures	ϵ-DF ↓	**0.205**	0.252	0.308	0.249	0.303	0.630	0.743
	γ-SF ↓	**0.003**	0.016	0.019	**0.003**	0.012	0.035	0.034
	p%-Rule ↑	**66.027**	65.278	64.276	63.853	62.296	63.096	60.062
	Bias Amp-DF ↓	**−0.333**	−0.286	−0.230	−0.289	−0.235	0.092	0.205
	Bias Amp-SF ↓	**−0.017**	−0.004	−0.001	**−0.017**	−0.008	0.015	0.014
**Adult Dataset**								
**Models**		**BDF-Classifier**			**BSF-Classifier**		**BPR-Classifier**	**BT-Classifier**
		$\epsilon_{1}=0.0$	$\epsilon_{1}=0.2231$	$\epsilon_{1}=\epsilon_{\mathrm{data}}$	$\gamma_{1}=0.0$	$\gamma_{1}=\gamma_{\mathrm{data}}$		
Tuning Parameter	λ	0.100	0.100	10.000	10.000	25.000	0.001	X
Performance Measures	Accuracy ↑	0.795	0.802	0.802	0.798	0.818	0.827	**0.831**
	F1 Score ↑	0.380	0.440	0.420	0.420	0.500	0.580	**0.600**
	ROC AUC ↑	0.836	0.847	0.860	0.835	0.876	0.876	**0.883**
Fairness Measures	ϵ-DF ↓	**0.305**	0.442	1.194	0.733	1.355	1.362	1.591
	γ-SF ↓	0.007	0.011	0.031	**0.001**	0.030	0.033	0.042
	p%-Rule ↑	55.590	54.855	46.988	**58.112**	47.128	54.071	45.019
	Bias Amp-DF ↓	**−1.075**	−0.938	−0.186	−0.647	−0.025	−0.018	0.211
	Bias Amp-SF ↓	−0.026	−0.022	−0.002	**−0.032**	−0.003	0.000	0.009

**Table 4 entropy-25-00660-t004:** Comparison of stochastic differential fair (SDF), stochastic subgroup fair (SSF), stochastic p%-Rule (SPR), and stochastic typical (ST) classifiers on the test set of HHP and HMDA datasets ($\epsilon_{1}=0.2231$ is the 80% rule). *Higher is better* for measures with ↑; *lower is better* for measures with ↓.

HHP Dataset								
**Models**		**SDF-Classifier**			**SSF-Classifier**		**SPR-Classifier**	**ST-Classifier**
		$\epsilon_{1}=0.0$	$\epsilon_{1}=0.2231$	$\epsilon_{1}=\epsilon_{\mathrm{data}}$	$\gamma_{1}=0.0$	$\gamma_{1}=\gamma_{\mathrm{data}}$		
Tuning Parameter	λ	0.900	0.900	1000.000	100.000	500.000	0.100	X
Performance Measures	Accuracy ↑	0.801	0.801	0.832	0.813	0.829	0.818	**0.843**
	F1 Score ↑	0.580	0.580	0.680	0.620	0.660	0.650	**0.720**
	ROC AUC ↑	0.796	0.803	0.859	0.870	0.887	0.873	**0.895**
Fairness Measures	ϵ-DF ↓	**0.210**	0.229	0.762	1.164	1.444	1.688	1.791
	γ-SF ↓	**0.004**	0.005	0.013	0.007	0.010	0.022	0.022
	p%-Rule ↑	61.507	61.292	56.183	53.678	51.169	**61.720**	49.860
	Bias Amp-DF ↓	**−1.250**	−1.231	−0.698	−0.296	−0.016	0.228	0.331
	Bias Amp-SF ↓	**−0.009**	−0.008	0.000	−0.006	−0.003	0.009	0.009
**HMDA Dataset**								
**Models**		**SDF-Classifier**			**SSF-Classifier**		**SPR-Classifier**	**ST-Classifier**
		$\epsilon_{1}=0.0$	$\epsilon_{1}=0.2231$	$\epsilon_{1}=\epsilon_{\mathrm{data}}$	$\gamma_{1}=0.0$	$\gamma_{1}=\gamma_{\mathrm{data}}$		
Tuning Parameter	λ	0.600	0.600	1.000	50.000	1000.000	0.010	X
Performance Measures	Accuracy ↑	0.819	0.821	0.848	0.851	0.858	0.858	**0.863**
	F1 Score ↑	0.890	0.890	0.900	0.900	**0.910**	**0.910**	**0.910**
	ROC AUC ↑	0.903	0.903	0.917	0.921	0.927	0.926	**0.931**
Fairness Measures	ϵ-DF ↓	**0.240**	0.296	0.417	0.436	0.549	0.554	0.673
	γ-SF ↓	0.004	0.005	0.007	**0.002**	0.005	0.008	0.011
	p%-Rule ↑	71.018	69.938	66.822	70.260	64.643	**71.256**	62.045
	Bias Amp-DF ↓	**−0.421**	−0.365	−0.244	−0.225	−0.112	−0.107	0.012
	Bias Amp-SF ↓	−0.005	−0.004	−0.002	**−0.007**	−0.004	−0.001	0.002

**Table 5 entropy-25-00660-t005:** Comparison of the inequity in the per-group allocation of the ϵ-DF and γ-SF metrics via the Gini coefficient.

	Gini Coefficient (*G*)			
Dataset	$G(\epsilon_{\mathrm{Data}})$	$G(\gamma_{\mathrm{Data}})$	$G(\epsilon_{\mathrm{LR}})$	$G(\gamma_{\mathrm{LR}})$
COMPAS	**0.151**	0.376	**0.135**	0.343
Adult	**0.099**	0.256	**0.126**	0.257
HHP	**0.113**	0.311	**0.105**	0.305
HMDA	**0.073**	0.423	**0.094**	0.358

**Table 6 entropy-25-00660-t006:** Protection of intersectionality property (Theorem 2). The cases in red are where γ-SF violates the intersectionality property enjoyed by ϵ-DF. Note that ϵ-DF and γ-SF are in different scales, so their values are not comparable. The purpose of this experiment is not to compare the values for ϵ-DF and γ-SF, but rather to empirically verify whether they respect the intersectionality property.

COMPAS Dataset					HHP Dataset		
**Protected Attributes**	ϵ **-DF**	γ **-SF**			**Protected Attributes**	ϵ **-DF**	γ **-SF**
race	0.1003	0.0070			gender	0.0505	0.0039
gender	0.9255	** 0.0656 **			age	2.0724	** 0.0469 **
race, gender	1.3156	0.0604			gender, age	2.2505	0.0241
**Adult Dataset**					**HMDA Dataset**		
**Protected Attributes**	ϵ **-DF**	γ **-SF**			**Protected Attributes**	ϵ **-DF**	γ **-SF**
nationality	0.2177	0.0045			ethnicity	0.1846	0.0056
race	0.9188	0.0128			race	0.6067	0.0126
gender	1.0266	** 0.0434 **			gender	0.1702	0.0088
gender, nationality	1.1511	0.0431			gender, ethnicity	0.3221	0.0118
race, nationality	1.1534	0.0163			race, ethnicity	0.6911	** 0.0167 **
race, gender	1.7511	0.0451			race, gender	0.6855	0.0137
race, gender, nationality	1.9751	0.0455			race, gender, ethnicity	0.8498	0.0163

## Data Availability

Publicly available datasets were analyzed in this study, except for the HDMA dataset. Restrictions apply to the availability of these data. The HDMA dataset was obtained from the Federal Financial Institutions Examination Council’s (FFIEC) and are available at https://www.ffiec.gov/hmda/ (accessed on 10 April 2023) with the permission of the FFIEC. Links and/or citations to the sources of the datasets used were provided in Section 10.1.

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
