# Peer review of "Differential Fairness: An Intersectional Framework for Fair AI†"

_entropy, 2023, doi:10.3390/e25040660_

Round 1

Reviewer 1 Report

This work addresses an important open challenge in algorithmic fairness - the question of how to measure model (un)fairness for people with respect to multiple sensitive attributes - by drawing on theories of intersectionality and methods from differential privacy. This is an ambitious paper, and I was impressed both by the authors’ careful and thoughtful approach, and by their deep commitment to interdisciplinarity. 

This paper contributes a theoretical definition for “differential fairness” (inspired by differential privacy); proofs of the validity of their approach and its improvement over other approaches; a learning algorithm to implement differential fairness; experiments on commonly used fairness datasets to demonstrate the validity; and a robust theoretical engagement with social science literature on intersectionality to inform their approach.

There is a lot to like about this paper, from the clear writing, the thoughtful interweaving of theories from social sciences and methods from differential privacy (and information theory more generally), to the empirical demonstration with “real-world” datasets. This is an exciting paper, which demonstrates the effectiveness of interdisciplinary approaches to algorithmic fairness. It will make a substantive contribution to practitioners working to implement fairness metrics and algorithms in their work, including via open-source fairness toolkits (e.g., Fairlearn, etc).

It is in that spirit that I offer some suggestions for the authors to revise to better sharpen the contributions of this paper and ideally maximize its impact across multiple interdisciplinary communities.

First, although I loved the robust theoretical engagement with social science theories of intersectionality in section 2, it came as a bit of a surprise to me, as the introduction did not feel like it was approaching the topic in as nuanced and interdisciplinary way as the rest of the paper. Specifically, lines 40-47, when contextualizing the definition of intersectionality with how it will be used in the rest of the paper, appeared to represent it as simply a matter of, e.g., tuples of multiple sensitive attributes, rather than, e.g., the effects of interlocking systems of oppression that manifest or impact people with those attributes. Section 2 has a quite nuanced treatment of the topic, so some revision of the intro would help interdisciplinary readers better appreciate what is to follow in the rest of the paper.

Secondly, there were some claims that stood out to me as seeming reductive in their approach to the sociotechnical nature of fairness, which was belied by the nuance of the rest of the paper, but which may mislead readers. For instance:

  • Lines 28-29: “...to the social goals which are to be achieved by mathematical fairness definitions” 

    • Mathematical fairness definitions will not achieve any social goals on their own, but will be used as part of larger sociotechnical systems.

  • Lines 53-78; 133-134; 300; 490-495: discussions of “provable guarantees on harms” 

    • I appreciate that this has a particular meaning in mathematical terms, but given that these approaches will (ideally) be used in real-world sociotechnical systems, for which there are no guarantees on harms, this feels like over-selling. More generally, while I appreciate the discussion of harms, it’s important not to conflate fairness metrics (for which there may be provable guarantees) with the harms those metrics are intended to measure (for which we cannot guarantee, when used in real-world systems). For more on this distinction, I suggest the authors read Jacobs and Wallach, 2021. 

  • Lines 67-76; 133-134; 490-495: discussions of how “stakeholders can interpret” the value of differential privacy or fairness, or how it “meaningfully represents” the construct of fairness.

    • Meaningful to whom? What evidence do the authors have for whether or how stakeholders can interpret the measure? There’s been a substantial literature on interpretability of fairness metrics, which, if this is something the authors are interested in claiming, they should better engage with and articulate. E.g., Wang et al., 2020; Cheng et al., 2019; Van Berkel et al., 2021, to name just a few.

Thirdly, I appreciated the diagram of the setting for the proposed differential fairness criterion (Figure 1), but this reveals one of the limitations that arose. That is, the differential fairness criterion is a measure of the fairness of the model, and not a measure of the fairness of the decisions made on the basis of that model. In other words, the “vendor” or user of the model in some decision-making system, either an algorithmic decision, or a sociotechnical system, may introduce their own biases into the decision process, or may neglect to adhere to the recommendations of the fair model. This is not a critique of the authors’ fairness criterion, but it is important to temper the claims for e.g., provable guarantees for harms, as a result. See Green and Chen, 2021; or Green, 2022 for more. This is also an issue in lines 205-212 as well.

I appreciated the footnote #2, but I would have liked this to be elevated into the main body of the text. Perhaps not with the framing as critiques raised for an earlier version of the paper, but the substantive points in this footnote were quite compelling and I would hate for less careful readers to gloss over them. 

Also, I know that different journals and disciplinary communities have different norms for structuring papers; however, I felt that including the related work at the end of the paper did a disservice to helping readers contextualize this paper. For instance, I found myself wondering throughout how this criteria related to other well-known fairness metrics, and while the authors do occasionally mention others (e.g., equalized odds, equality of opportunity), it wasn’t until the very end of the paper that we got some treatment of those. This may be better earlier on (at least in some, perhaps abbreviated, form).

A more minor point: in Figure 2b, how might the authors account for situations where some attributes may result in privilege and others where the systems of oppression may be more salient? Or, for attributes where people may be able to “pass” as e.g., another race, or sexual orientation. This is not a major concern by any means, but may be worth thinking about as future extensions. 

Other misc questions/comments: 

  • Typo: line 173: “policy” → “police”?

  • Line 406: “more conservative” than what?

  • Line 416: what is a “legitimate source of inequity”? What does that mean?

  • Line 528-530: How do you know that the number of intersectional groups is in that range?

  • Line 833: Does the entire population really “have a voice” in the determination of (un)fairness? It’s important not to oversell. This paper does so much, so well, that spurious claims like this are not necessary and undermine the wonderful contributions of the rest of this work.

  • Cheng, H. F., Wang, R., Zhang, Z., O'Connell, F., Gray, T., Harper, F. M., & Zhu, H. (2019, May). Explaining decision-making algorithms through UI: Strategies to help non-expert stakeholders. In Proceedings of the 2019 chi conference on human factors in computing systems (pp. 1-12).

  • Green, B., & Chen, Y. (2021). Algorithmic risk assessments can alter human decision-making processes in high-stakes government contexts. Proceedings of the ACM on Human-Computer Interaction, 5(CSCW2), 1-33.

  • Green, B. (2022). The flaws of policies requiring human oversight of government algorithms. Computer Law & Security Review, 45, 105681.

  • Jacobs, A. Z., & Wallach, H. (2021, March). Measurement and fairness. In Proceedings of the 2021 ACM conference on fairness, accountability, and transparency (pp. 375-385).

  • Wang, R., Harper, F. M., & Zhu, H. (2020, April). Factors influencing perceived fairness in algorithmic decision-making: Algorithm outcomes, development procedures, and individual differences. In Proceedings of the 2020 CHI Conference on Human Factors in Computing Systems (pp. 1-14).

  • Van Berkel, N., Goncalves, J., Russo, D., Hosio, S., & Skov, M. B. (2021, May). Effect of information presentation on fairness perceptions of machine learning predictors. In Proceedings of the 2021 CHI Conference on Human Factors in Computing Systems (pp. 1-13).

Author Response

Please see the attachment, specifically the section marked "Reviewer 1."

Reviewer 2 Report

This paper presents a fairness definition that focuses on the intersectionality of different protective attributes. The definition tries to cover multiple aspects and views of fairness, inspired by other fields and research topics (privacy, philosophy, etc). The experimental results evaluate different important aspects of the proposed algorithms. 

Overall, the problem is very important. The authors have followed a holistic approach while considering this problem. The computational challenges were also considered, that's why two different versions of the algorithm were presented (batch and stochastic differential fairness models). The results are also nicely organized into different sections. Different datasets with various sizes from various applications have also been tested. The paper is carefully written. Sect. 5 with the example was very useful.

The limitations of the paper mostly come from the presentation of the paper. The first sections do not follow the structure/flow of a typical research paper. Some phrases are very general about the problem. The first pages focus on the motivation of the paper, and how the authors came up with this approach. Little emphasis is given to the gap in existing research, or how this model compares to different research approaches. 

Also, there is a bit controversial point in the criteria that the authors propose for an intersectional definition of fairness. In particular, criterion D: "The definition should protect minority groups". That is something that might not necessarily be done by a definition, i.e., "protect". Maybe that can be rephrased.

Some suggestions:

- to make the paper cleaner to read, all interdisciplinary discussions/considerations could be moved together.

- possibly, a short subsection that discusses any limitations/assumptions of the work could be beneficial. For example, this work requires us to know the values of the protected attributes.

Overall, the paper could be shorter and easier for the readers to go through. There are some repetitive points (for example, the definition of differential fairness is appearing twice in the paper (2 and 4)). The paper needs a lot of math, so the notation should be a bit clearer. 

Author Response

Please see the attachment, specifically the section marked "Reviewer 2."

Reviewer 3 Report

The differential privacy idea is very interesting and I believe the paper writin is excellent. However, I have a huge concern about if it is an original research article. 

I just had a quick search and found that there are a lot of similar content written by the same authors. 

For example, a workshop paper, Differential Privacy (Foulds, James R.; Islam, Rashidul; Keya, Kamrun Naher; Pan, Shimei), published on NeurIPS 2019 Workshop on Machine Learning with Guarantees.

Also, a conference paper with official DOI (Foulds, James R., et al. "An intersectional definition of fairness." 2020 IEEE 36th International Conference on Data Engineering (ICDE). IEEE, 2020, DOI 10.1109/ICDE48307.2020.00203).

Just quickly going through the full version (stored in arXiv) of the above conference paper, I cannot find significant difference between full version conference paper and this submission.  There are some slight differences, for example, in the old one, there are only two experiments, but this new submission has the experiments on four datasets. 

Therefore, I don't think these two can be considered as two original research papers.

Author Response

Please see the attachment, specifically the section marked "Reviewer 3."

Round 2

Reviewer 1 Report

I want to thank the authors for their thoughtful and detailed revisions to the draft. The authors have diligently responded to all of my concerns, and I am satisfied with the current state of the draft. This is an impressive example of interdisciplinary work and will, I suspect, be an exemplary model for many future readers. I look forward to sharing and citing this paper in the future. 

Reviewer 3 Report

After reviewing the article and considering the revisions made by the author, I am confident that the concerns about originality have been satisfactorily clarified. Furthermore, the revisions have addressed most of the concerns raised by the other reviewers. Therefore, I recommend that the current version of the article be published in Entropy.